# BAYESIAN REGULARIZATION OF LATENT REPRESENTATION

Chukwudi Paul Obite    Zhi Chang    Keyan Wu    Shiwei Lan*

*School of Mathematical & Statistical Sciences, Arizona State University*
*901 S Palm Walk, Tempe, AZ 85287, USA*

## ABSTRACT

The effectiveness of statistical and machine learning methods depends on how well data features are characterized. Developing informative and interpretable latent representations with controlled complexity is essential for visualizing data structure and for facilitating efficient model building through dimensionality reduction. Latent variable models, such as Gaussian Process Latent Variable Models (GP-LVM), have become popular for learning complex, nonlinear representations as alternatives to Principal Component Analysis (PCA). In this paper, we propose a novel class of latent variable models based on the recently introduced Q-exponential process (QEP), which generalizes GP-LVM with a tunable complexity parameter, $q > 0$. Our approach, the *Q-exponential Process Latent Variable Model (QEP-LVM)*, subsumes GP-LVM as a special case when $q = 2$, offering greater flexibility in managing representation complexity while enhancing interpretability. To ensure scalability, we incorporate sparse variational inference within a Bayesian training framework. We establish connections between QEP-LVM and probabilistic PCA, demonstrating its superior performance through experiments on datasets such as the Swiss roll, oil flow, and handwritten digits.

**Keywords:** Dimensionality Reduction, Latent Variable Models, Representation Complexity Regularization, Variational Inference, Generative Models.

## 1 INTRODUCTION

In various learning tasks involving high-dimensional data, it is crucial to effectively learn a representation of the data that lies in a low-dimensional subspace. Such latent representation is often the key in unsupervised learning to understand the complex data structure which is infeasible to visualize in the original space. What is more, reducing data to low-dimensional latent space is also the core to build efficient models in supervised learning. Principal component analysis (PCA) (Pearson, 1901; Jolliffe, 1986; Jolliffe & Cadima, 2016) has long been used as a classic technique in factor analysis. It is later cast into a probabilistic latent variable model (LVM) with a linear (Tipping & Bishop, 1999; Minka, 2000) or a nonlinear (Schölkopf et al., 1998) mapping from the latent space to the data feature space.

In the probabilistic PCA literature, Lawrence (2003; 2005) propose to employ Gaussian process (GP) (Rasmussen & Williams, 2005) as a prior for the mapping in LVM. Such GP-LVM views data as functional outputs of latent variables and can be regarded as a linear or nonlinear probabilistic PCA through a linear or more general kernel respectively. Titsias & Lawrence (2010) then develop a Bayesian version of GP-LVM and an efficient variational Bayes inference with sparse approximation through inducing points (Titsias, 2009). Since then it has been extensively applied in neuroscience (Gundersen et al., 2021), bioinformatics (Ahmed et al., 2018), and robotics (Delgado-Guerrero et al., 2020), etc. More recent development extends GP-LVM to discriminative classification (Urtasun & Darrell, 2007), Gaussian mixture models (Nickisch & Rasmussen, 2010), deep probabilistic models (Damianou & Lawrence, 2013), inverse problems (Atkinson & Zabaras, 2019), and longitudinal modeling (Le & Honavar, 2020).

---

*slan@asu.edu

All the aforementioned works are based on GP which tends to be over-smooth for inhomogeneous data with abrupt changes or sharp contrast. To address this issue, a new stochastic process named $Q$-Exponential process (Q-EP) (Li et al., 2023) has been proposed based on $L_q$ penalty to regularize the modeling effect through a parameter $q > 0$ which also embraces GP as a special case for $q = 2$. In this paper, we aim to port such flexibility in regularization to LVM. More specifically, we replaces GP in GP-LVM (Lawrence, 2003) with Q-EP to propose a novel class of LVMs named *Q-exponential Process Latent Variable Models (QEP-LVM)* parameterized by the regularization parameter $q > 0$. We establish its connection to the probabilistic PCA and provide explicit formula for the maximum likelihood estimator of the latent variable singular values. We also develop the Bayesian QEP-LVM and adopt variational inference as Titsias & Lawrence (2010) with sparse approximation via inducing points (Titsias, 2009). To derive the evidence lower bound (ELBO) for the variational form of Bayesian QEP-LVM, we adopt the approach by Hensman et al. (2015) to directly compute the variational distribution of latent function instead of using variational calculus (Titsias & Lawrence, 2010). Though not straightforward in the Q-EP setting, tractable form of ELBO can be obtained with the help of Jensen's inequality.

We demonstrate the regularization effect through the parameter $q > 0$ on the latent representations learned with QEP-LVM some of which exhibit controlled compactness or enhanced interpretability compared with GP-LVM ($q = 2$) (See Section 4). It mains as an open question to determine the optimal $q^*$ automatically based on the given data. We therefore adopt a Bayesian approach by imposing appropriate priors on $q$ to obtain an optimal choice. With chosen regularization parameters, we also investigate the generative classification models built on fitted LVMs and show that GP-LVM is often sub-optimal, especially for inhomogeneous data. All the numerical examples have been efficiently implemented in `GPyTorch` (Gardner et al., 2018).

**Connection to existing works on non-Gaussian LVMs** Our proposed QEP-LVM directly generalizes GP-LVM (Titsias & Lawrence, 2010) with flexible regularization on the learned latent representations. There are also some works regarding non-Gaussian LVMs falling in the same category of QEP-LVM. Palmer et al. (2005) theoretically characterize variational representation of LVMs with non-Gaussian priors constructed as a supremum or an integral over the scale parameter Gaussian densities. Kleppe & Skaug (2008) build and fit non-Gaussian LVMs via the moment-generating function. Salehkaleybar et al. (2020); Xie et al. (2022); Chen et al. (2022) consider linear causal LVMs with non-Gaussian noises. Gundersen et al. (2021); Zhang et al. (2023); Li et al. (2024) propose non-Gaussian LVMs with random Fourier features in the Karhunen-Loéve expansion of GP tailored to non-Gaussian data. The non-Gaussianity of these LVMs is constructed by certain composition of GPs. On the contrary, the QEP-LVM is, if not the first, developed directly based on a non-Gaussian stochastic process. Our proposed work has multi-fold contributions to the literature of LVM:

1. We propose a novel LVM based on Q-EP which learns latent data representation with flexible complexity.

2. We establish the connection between QEP-LVM and nonlinear probabilistic PCA.

3. We demonstrate the regularization effect of Bayesian QEP-LVM which enables enhanced latent representation learning compared with GP-LVM.

The rest of the paper is organized as follows. Section 2 reviews Q-EP as a flexible prior for Bayesian multi-output regression. We then introduce QEP-LVM as a nonlinear probabilistic PCA and the variational inference for Bayesian QEP-LVM in Section 3. Section 4 investigates the regularization effect of QEP-LVM via parameter $q > 0$ and demonstrate that the optimal $q^*$ leads to better latent representation compared with that obtained by GP-LVM ($q = 2$). Finally, we conclude in Section 5 with discussion on some future directions.

## 2 BACKGROUND: BAYESIAN MODELS WITH Q-EXPONENTIAL PRIORS

### 2.1 $Q$-EXPONENTIAL DISTRIBUTION AND PROCESS

Motivated by $L_q$ regularization, Li et al. (2023) propose the *multivariate q-exponential distribution* that is exchangable and consistent with respect to marginalization.

**Definition 1.** *A multivariate q-exponential distribution for a random vector* $\mathbf{u} \in \mathbb{R}^N$, *denoted as* q-*$ED_N(\boldsymbol{\mu}, \mathbf{C})$, has the following density*

$$p(\mathbf{u}|\boldsymbol{\mu}, \mathbf{C}, q) = \frac{q}{2}(2\pi)^{-\frac{N}{2}}|\mathbf{C}|^{-\frac{1}{2}}r^{(\frac{q}{2}-1)\frac{N}{2}}\exp\left\{-\frac{r^{\frac{q}{2}}}{2}\right\}, \quad r(\mathbf{u}) = (\mathbf{u}-\boldsymbol{\mu})^\mathsf{T}\mathbf{C}^{-1}(\mathbf{u}-\boldsymbol{\mu}).$$

**Remark 1.** *The above density (log-convex, heavy-tailed around mean for $0 < q < 2$ and log-concave, light-tailed for $q > 2$), when taken negative logarithm, is dominated by some weighted $L_q$ norm of $\mathbf{u} - \boldsymbol{\mu}$, i.e. $\frac{1}{2}r^{\frac{q}{2}} = \frac{1}{2}\|\mathbf{u} - \boldsymbol{\mu}\|_{\mathbf{C}}^q$. From the optimization perspective, q-$ED_N$, when used as a prior, imposes $L_q$ regularization in obtaining the maximum a posteriori (MAP).*

Suppose a function $u(x)$ is observed at $N$ locations, $x_1, \cdots, x_N \in \mathcal{D} \subset \mathbb{R}^d$. Li et al. (2023) define the following *q-exponential process (Q-EP)* based on a scaled *q*-exponential distribution q-$ED_N^*(\mathbf{0}, \mathbf{C})$ assumed for the random vector $\mathbf{u} = (u(x_1), \cdots, u(x_N))$.

**Definition 2.** *A (centered) q-exponential process $u(x)$ with kernel $\mathcal{C}$, q-$\mathcal{EP}(0, \mathcal{C})$, is a collection of random variables such that any finite set, $\mathbf{u} := (u(x_1), \cdots u(x_N))$, follows a scaled multivariate q-exponential distribution q-$ED^*(\mathbf{0}, \mathbf{C})$, i.e., $N^{-\frac{1}{2}+\frac{1}{q}}\mathbf{u} \sim$ q-$ED_N(\mathbf{0}, \mathbf{C})$, where $\mathbf{C} = [\mathcal{C}(x_i, x_j)]_{N \times N}$. If $\mathcal{C} = \mathcal{I}$, then $u$ is said to be* marginally identical but uncorrelated (m.i.u.).

**Remark 2.** *If $q = 2$, q-$ED_N(\boldsymbol{\mu}, \mathbf{C})$ reduces to $\mathcal{N}_N(\boldsymbol{\mu}, \mathbf{C})$ and q-$\mathcal{EP}(0, \mathcal{C})$ becomes $\mathcal{GP}(0, \mathcal{C})$. When $q \in [1, 2)$, q-$\mathcal{EP}(0, \mathcal{C})$ lends flexibility to modeling functional data with more regularization than GP (See Figure 1 for the regularization effect of q).*

For multiple Q-EPs, $(u_1(x), \cdots, u_D(x))$, we define multi-output (multivariate) Q-EP through matrix vectorization, $\text{vec}(\mathbf{U}_{N \times D}) = [\mathbf{u}_1^\mathsf{T}, \cdots, \mathbf{u}_D^\mathsf{T}]^\mathsf{T}$, which forms a vector by concatenating its columns. Note, the component processes are often assumed uncorrelated, rather than independent.

**Definition 3.** *A multi-output (multivariate) q-exponential process, $u(\cdot) = (u_1(\cdot), \cdots, u_D(\cdot))$, each $u_j(\cdot) \sim$ q-$\mathcal{EP}(\mu_j, \mathcal{C}_x)$, is said to have association $\mathbf{C}_t$ if at any finite locations, $\mathbf{x} = \{x_n\}_{n=1}^N$, $\text{vec}([u_1(\mathbf{x}), \cdots, u_D(\mathbf{x})]_{N \times D}) \sim$ q-$ED_{ND}(\text{vec}(\boldsymbol{\mu}), \mathbf{C}_t \otimes \mathbf{C}_x)$, where we have $u_j(\mathbf{x}) = [u_j(x_1), \cdots, u_j(x_N)]^\mathsf{T}$, for $j = 1, \ldots, D$, $\boldsymbol{\mu} = [\mu_1(\mathbf{x}), \cdots, \mu_D(\mathbf{x})]_{N \times D}$ and $\mathbf{C}_x = [\mathcal{C}_x(x_n, x_m)]_{N \times N}$. We denote $u \sim$ q-$\mathcal{EP}(\mu, \mathcal{C}_x, \mathbf{C}_t)$. In particular, the component processes are m.i.u. if $\mathbf{C}_t = \mathbf{I}_D$.*

## 2.2 BAYESIAN REGRESSION WITH Q-EP PRIORS

Given data $\mathbf{x} = \{x_n\}_{n=1}^N$ and $\mathbf{y} = \{y_n\}_{n=1}^N$, for the generic Bayesian regression model:

$$\begin{aligned}
\mathbf{y} &= f(\mathbf{x}) + \boldsymbol{\varepsilon}, \quad \boldsymbol{\varepsilon} \sim \text{q-}ED_N(0, \Sigma), \\
f &\sim \text{q-}\mathcal{EP}(0, \mathcal{C}),
\end{aligned} \tag{1}$$

we have a tractable posterior (predictive) distribution similar to GP regression as in Theorem 3.5 of Li et al. (2023).

Denote $\mathbf{X} = [\mathbf{x}_1, \cdots, \mathbf{x}_Q]_{N \times Q}$, $\mathbf{F} = [f_1(\mathbf{X}), \cdots, f_D(\mathbf{X})]_{N \times D}$ and $\mathbf{Y} = [\mathbf{y}_1, \cdots, \mathbf{y}_D]_{N \times D}$. With m.i.u. Q-EP priors as in Definition 3 imposed on $f := (f_1, \cdots, f_D)$, now we consider the following multivariate regression problem:

$$\begin{aligned}
\text{likelihood}: \quad &\text{vec}(\mathbf{Y})|\mathbf{F} \sim \text{q-}ED_{ND}(\text{vec}(\mathbf{F}), \mathbf{I}_D \otimes \Sigma), \\
\text{prior on latent function}: \quad &f \sim \text{q-}\mathcal{EP}(0, \mathcal{C}, \mathbf{I}_D).
\end{aligned} \tag{2}$$

Noticing that $\mathbf{Y} = \mathbf{F} + \boldsymbol{\varepsilon}$ with $\text{vec}(\boldsymbol{\varepsilon}) \sim \text{q-}ED(\mathbf{0}, \mathbf{I}_D \otimes \Sigma)$, we can find the marginal of $\mathbf{Y}$ based on the property of q-ED (Fang & Zhang, 1990) as follows:

$$\text{marginal likelihood}: \quad \text{vec}(\mathbf{Y})|\mathbf{X} \sim \text{q-}ED_{ND}(\mathbf{0}, \mathbf{I}_D \otimes (\mathbf{C} + \Sigma)). \tag{3}$$

In the following, we view $\mathbf{Y}$ as the only *data* and $\mathbf{X}$ is merely regarded as the *latent variable*. Usually it is assumed $Q \ll D$ in the latent representation learning. Different from standard regression problems, a latent variable model (LVM) seeks to solve $\mathbf{X}$ in (2) or equivalently (3) for given data $\mathbf{Y}$ either deterministically or probabilistically (assuming a proper prior on $\mathbf{X}$).

## 3 Q-EP LATENT VARIABLE MODEL

GP-LVM is introduced by Lawrence (2003; 2005) as an unsupervised learning method for dimensionality reduction. It can be interpreted as a nonlinear extension of probabilistic PCA (Tipping & Bishop, 1999; Minka, 2000). The GP can be replaced by Q-EP to impose more regularization on the latent space, and hence we propose a Q-EP based LVM (QEP-LVM).

### 3.1 QEP-LVM AS A NON-LINEAR PROBABILISTIC PCA

For convenience, we set $\Sigma = \beta^{-1}\mathbf{I}_N$ in the following. Let us start with linear latent function $\mathbf{F} = \mathbf{X}\mathbf{W}$ in the model (2) with $\mathbf{W} \in \mathbb{R}^{Q \times D}$. The original probabilistic PCA (Tipping & Bishop, 1999) is formulated by integrating out the latent variable $\mathbf{X}$: $p(\mathbf{Y}|\mathbf{W}, \beta) = \int p(\mathbf{Y}|\mathbf{X}, \mathbf{W}, \beta)p(\mathbf{X})d\mathbf{X}$ with prior $p(\mathbf{X}) = \mathcal{N}(\mathbf{X}; \mathbf{0}, \mathbf{I}_{NQ})$. The resulting marginal likelihood, $p(\mathbf{Y}|\mathbf{W}, \beta) = \mathcal{N}(\mathbf{Y}; \mathbf{0}, (\mathbf{W}^\mathsf{T}\mathbf{W} + \beta^{-1}\mathbf{I}_D) \otimes \mathbf{I}_N)$, is maximized for the solution $\mathbf{W}^*$, which spans the principal space of the data $\mathbf{Y}$.

Lawrence (2003) forms the dual problem of probabilistic PCA by marginalizing the weight parameter $\mathbf{W}$ as $p(\mathbf{Y}|\mathbf{X}, \beta) = \int p(\mathbf{Y}|\mathbf{X}, \mathbf{W}, \beta)p(\mathbf{W})d\mathbf{W}$ with prior $p(\mathbf{W}) = \mathcal{N}(\mathbf{W}; \mathbf{0}, \mathbf{I}_{QD})$ and optimizing the resulting $p(\mathbf{Y}|\mathbf{X}, \beta) = \mathcal{N}(\mathbf{Y}; \mathbf{0}, \mathbf{I}_D \otimes (\mathbf{X}\mathbf{X}^\mathsf{T} + \beta^{-1}\mathbf{I}_N))$ for the optimum, $\mathbf{X}^*$, which is related to the eigen-decomposition of empirical covariance matrix $\mathbf{Y}\mathbf{Y}^\mathsf{T}$. Lawrence (2003; 2005) replace the above linear kernel with general ones, e.g. $\mathbf{K} = \mathbf{C} + \Sigma$ as in (3), to learn a nonlinear latent reduction in $\mathbf{X}$, essentially defining an LVM with GP mapping (GP-LVM).

Now we impose a m.i.u. Q-EP prior on $\mathrm{vec}(\mathbf{W})|\alpha \sim \text{q-ED}(\mathbf{0}, \alpha^{-1}\mathbf{I}_{QD})$ that induces the following prior on the latent function $\mathbf{F}$:

$$\mathrm{vec}(\mathbf{F}) = (\mathbf{I}_D \otimes \mathbf{X})\mathrm{vec}(\mathbf{W}) \sim \text{q-ED}(0, \alpha^{-1}\mathbf{I}_D \otimes \mathbf{X}\mathbf{X}^\mathsf{T}).$$

The following marginal likelihood of $\mathbf{Y}|\mathbf{X}$ can be obtained similar to (3), which defines a stochastic mapping from the latent space of $\mathbf{X}$ to the data space of $\mathbf{Y}$:

$$\mathrm{vec}(\mathbf{Y})|\mathbf{X}, \alpha, \beta \sim \text{q-ED}(\mathbf{0}, \mathbf{I}_D \otimes \mathbf{K}), \quad \mathbf{K} = \alpha^{-1}\mathbf{X}\mathbf{X}^\mathsf{T} + \beta^{-1}\mathbf{I}_N. \tag{4}$$

Similarly as GP-LVM, (3), or (4) with general kernel $\mathbf{K}$ defines an LVM with Q-EP mapping (QEP-LVM). Let $r(\mathbf{Y}) = \mathrm{vec}(\mathbf{Y})^\mathsf{T}(\mathbf{I}_D \otimes \mathbf{K})^{-1}\mathrm{vec}(\mathbf{Y}) = \mathrm{tr}(\mathbf{K}^{-1}\mathbf{Y}\mathbf{Y}^\mathsf{T})$. The log-likelihood of (4) is

$$L = -\frac{D}{2}\log|\mathbf{K}| + \frac{ND}{2}\left(\frac{q}{2} - 1\right)\log r(\mathbf{Y}) - \frac{1}{2}r^{\frac{q}{2}}(\mathbf{Y}). \tag{5}$$

The following theorem states that the maximum likelihood estimator (MLE) for $\mathbf{X}$ is equivalent to the solution for the dual probabilistic PCA (Tipping & Bishop, 1999; Minka, 2000).

**Theorem 3.1.** *Suppose $\mathbf{Y}\mathbf{Y}^\mathsf{T}/D$ has eigen-decomposition $\mathbf{U}\mathbf{\Lambda}\mathbf{U}^\mathsf{T}$ with $\mathbf{\Lambda}$ being the diagonal matrix with eigenvalues $\{\lambda_i\}_{i=1}^N$. Then the MLE for (5) is*

$$\mathbf{X}^* = \mathbf{U}_Q\mathbf{L}\mathbf{V}, \ \mathbf{L} = \mathrm{diag}(\{\sqrt{\alpha(c\lambda_i - \beta^{-1})}\}_{i=1}^Q), \ c(q) = D^{1-\frac{2}{q}}(D \wedge Q)\left[\frac{q}{2(D \wedge Q) + (q-2)N}\right]^{\frac{2}{q}},$$

*where $\mathbf{U}_Q$ is an $N \times Q$ matrix with the first $Q$ eigen-vectors in $\mathbf{U}$, $\mathbf{V}$ is an arbitrary $Q \times Q$ orthogonal matrix, and $a \wedge b := \min\{a, b\}$.*

*Proof.* See Appendix A. $\square$

**Remark 3.** *When $q = 2$, the singular values of the latent variable $\mathbf{X}$ reduce to $l_i = \sqrt{\alpha(\lambda_i - \beta^{-1})}$ corresponding to those for GP-LVM, though they were mistakenly stated as $l_i = (\alpha^{-1}(\lambda_i - \beta^{-1}))^{-\frac{1}{2}}$ in Lawrence (2003). When $q > 0$ varies, it regularizes the singular values (hence the latent space spaced by $\mathbf{X}$) through $c(q)$, whose properties are illustrated in Figure A.1.*

To detect complicated nonlinear latent representation, we consider the following automatic relevance determination (ARD) squared exponential (SE) kernel $k(\cdot, \cdot)$ (Titsias & Lawrence, 2010):

$$\mathbf{K} = [k(\mathbf{x}_n, \mathbf{x}_m)]_{N \times N}, \quad k(\mathbf{x}_n, \mathbf{x}_m) = \alpha^{-1}\exp\left\{-\frac{1}{2}(\mathbf{x}_n - \mathbf{x}_m)^\mathsf{T}\mathrm{diag}(\boldsymbol{\gamma})(\mathbf{x}_n - \mathbf{x}_m)\right\}. \tag{6}$$

Then the locale of $\mathbf{X}$ can be determined by optimizing (5) with respect to $\mathbf{X}$. The full solution to QEP-LVM also involves optimizing the kernel parameters $(\alpha, \boldsymbol{\gamma})$. Other kernels may lead to slightly different results, but the numerical conclusion (Section 4) is robust to the choice of kernels.

## 3.2 BAYESIAN QEP-LVM

In this section we introduce Bayesian QEP-LVM and develop variational inference as Bayesian GP-LVM (Titsias & Lawrence, 2010). Compared with the optimization method (Lawrence, 2003), the Bayesian training procedure is robust to over-fitting and can automatically determine the dimensionality of the nonlinear latent space (Titsias & Lawrence, 2010) e.g. by thresholding the correlation length $\gamma$. The derivation of variational Bayes for QEP-LVM is much more involved because the log-likelihood (5) is no longer presented as a quadratic form of data. Yet an appropriate evidence lower bound (ELBO) can still be obtained with Jensen's inequality.

In addition to the likelihood model (4), we imposes a prior on the latent variable $\mathbf{X}$ and consider the following Bayesian QEP-LVM:

$$\text{marginal likelihood}: \quad \text{vec}(\mathbf{Y})|\mathbf{X} \sim \text{q-ED}(\mathbf{0}, \mathbf{I}_D \otimes \mathbf{K}),$$
$$\text{prior on latent variable}: \quad \text{vec}(\mathbf{X}) \sim \text{q-ED}(\mathbf{0}, \mathbf{I}_{NQ}).$$

We use variational Bayes to approximate the posterior distribution $p(\mathbf{X}|\mathbf{Y}) \propto p(\mathbf{Y}|\mathbf{X})p(\mathbf{X})$ with the variational distribution using an uncorrelated q-ED:

$$\text{variational distribution for latent variable}: \quad q(\mathbf{X}) \sim \text{q-ED}(\boldsymbol{\mu}, \text{diag}(\{\mathbf{S}_n\})),$$

where each covariance $\mathbf{S}_n$ is of size $Q \times Q$ and can be chosen as a diagonal matrix for convenience. Due to the equality of log-evidence, $\log p(\mathbf{Y}) = \text{KL}(q(\mathbf{X})\|p(\mathbf{X}|\mathbf{Y})) + \mathcal{L}(q(\mathbf{X}))$, minimizing the KL divergence is equivalent to maximizing the ELBO $\mathcal{L}(q(\mathbf{X}))$ as follows:

$$\log p(\mathbf{Y}) \geq \mathcal{L}(q(\mathbf{X})) := \int q(\mathbf{X}) \log \frac{p(\mathbf{Y}|\mathbf{X})p(\mathbf{X})}{q(\mathbf{X})} d\mathbf{X} = \tilde{\mathcal{L}}(q(\mathbf{X})) - \text{KL}(q(\mathbf{X})\|p(\mathbf{X})),$$

where the first term $\tilde{\mathcal{L}}(q(\mathbf{X})) = \int q(\mathbf{X}) \log p(\mathbf{Y}|\mathbf{X}) d\mathbf{X} =: \langle \log p(\mathbf{Y}|\mathbf{X}) \rangle_{q(\mathbf{X})}$ is intractable and hence difficult to bound.

### 3.2.1 LOWER BOUND FOR THE MARGINAL LIKELIHOOD

To address such intractability issue and to speed up the computation, sparse variational approximation (Titsias, 2009) is adopted by introducing a set of inducing points $\tilde{\mathbf{X}} \in \mathbb{R}^{M \times Q}$ with their function values $\mathbf{U} = [f_1(\tilde{\mathbf{X}}), \cdots, f_D(\tilde{\mathbf{X}})] \in \mathbb{R}^{M \times D}$. Hence the marginal likelihood $p(\mathbf{Y}|\mathbf{X})$ defined in (5) can be augmented to the following joint distribution each being a q-ED:

$$p(\mathbf{Y}|\mathbf{X}) \propto p(\mathbf{Y}|\mathbf{F})p(\mathbf{F}|\mathbf{U}, \mathbf{X}, \tilde{\mathbf{X}})p(\mathbf{U}|\tilde{\mathbf{X}}),$$

where we have $\text{vec}(\mathbf{U})|\tilde{\mathbf{X}} \sim \text{q-ED}(\mathbf{0}, \mathbf{I}_D \otimes \mathbf{K}_{MM})$ and the conditional distribution

$$\text{vec}(\mathbf{F})|\mathbf{U}, \mathbf{X}, \tilde{\mathbf{X}} \sim \text{q-ED}(\text{vec}(\mathbf{K}_{NM}\mathbf{K}_{MM}^{-1}\mathbf{U}), \mathbf{I}_D \otimes (\mathbf{K}_{NN} - \mathbf{K}_{NM}\mathbf{K}_{MM}^{-1}\mathbf{K}_{MN})). \quad (7)$$

The inducing points $\tilde{\mathbf{X}}$ are regarded as variational parameters and hence they are dropped from the following probability expressions. We then approximate $p(\mathbf{F}, \mathbf{U}|\mathbf{X}) \propto p(\mathbf{F}|\mathbf{U}, \mathbf{X})p(\mathbf{U})$ with $q(\mathbf{F}, \mathbf{U}) = p(\mathbf{F}|\mathbf{U}, \mathbf{X})q(\mathbf{U})$ in another variational Bayes as follows:

$$\log p(\mathbf{Y}|\mathbf{X}) \geq \int q(\mathbf{F}, \mathbf{U}) \log \frac{p(\mathbf{Y}|\mathbf{F})p(\mathbf{F}|\mathbf{U}, \mathbf{X})p(\mathbf{U})}{q(\mathbf{F}, \mathbf{U})} d\mathbf{F} d\mathbf{U}$$
$$= \int p(\mathbf{F}|\mathbf{U})q(\mathbf{U})d\mathbf{U} \log p(\mathbf{Y}|\mathbf{F})d\mathbf{F} + \int q(\mathbf{U}) \log \frac{p(\mathbf{U})}{q(\mathbf{U})} d\mathbf{U}. \quad (8)$$

Different from Titsias (2009); Titsias & Lawrence (2010) using the variational calculus, (SVGP Hensman et al., 2015) compute the marginal likelihood ELBO (8) through the variational distribution of latent function $\mathbf{F}$. Instead of the variational free form, we follow Hensman et al. (2015) to use the variational distribution for $\mathbf{U}$ of the following format conjugate to $p(\mathbf{F}|\mathbf{U})$:

$$\text{variational distribution for inducing values}: \quad q(\mathbf{U}) \sim \text{q-ED}(\mathbf{M}, \text{diag}(\{\boldsymbol{\Sigma}_d\})),$$

where $\mathbf{M}$ is the variational mean. Noticing that $\mathbf{F}|\mathbf{U}$ follows a conditional $q$-exponential (7), we can obtain the variational distribution of $\mathbf{F}$, $q(\mathbf{F})$, by marginalizing $\mathbf{U}$ out similarly as in (3):

$$\text{variational distribution for latent function}: q(\mathbf{F}) = \int q(\mathbf{F}, \mathbf{U})d\mathbf{U} = \int p(\mathbf{F}|\mathbf{U})q(\mathbf{U})d\mathbf{U} \sim \text{q-ED}$$

$$(\text{vec}(\mathbf{K}_{NM}\mathbf{K}_{MM}^{-1}\mathbf{M}), \mathbf{I}_D \otimes (\mathbf{K}_{NN} - \mathbf{K}_{NM}\mathbf{K}_{MM}^{-1}\mathbf{K}_{MN}) + \text{diag}(\{\mathbf{K}_{NM}\mathbf{K}_{MM}^{-1}\boldsymbol{\Sigma}_d\mathbf{K}_{MM}^{-1}\mathbf{K}_{MN}\})).$$

Therefore, the variational lower bound of the marginal likelihood (8) becomes

$$\log p(\mathbf{Y}|\mathbf{X}) \geq \langle \log p(\mathbf{Y}|\mathbf{F}) \rangle_{q(\mathbf{F})} - \mathrm{KL}(q(\mathbf{U})\|p(\mathbf{U})).$$

Denote by $\log p(\mathbf{Y}|\mathbf{F}) = \varphi_0(r(\mathbf{Y}, \mathbf{F}))$, where $\varphi_0(r) := \frac{DN}{2}\log\beta + \frac{ND}{2}\left(\frac{q}{2}-1\right)\log r - \frac{1}{2}r^{\frac{q}{2}}$ is convex for $q \in (0, 2]$, and $r(\mathbf{Y}, \mathbf{F}) = \mathrm{vec}(\mathbf{Y} - \mathbf{F})^{\mathsf{T}}(\beta^{-1}\mathbf{I}_{ND})^{-1}\mathrm{vec}(\mathbf{Y} - \mathbf{F}) = \beta\mathrm{tr}((\mathbf{Y} - \mathbf{F})(\mathbf{Y} - \mathbf{F})^{\mathsf{T}})$ is a quadratic form of random variable $\mathbf{Y}$. Therefore, by Jensen's inequality, we can bound from below

$$\langle \log p(\mathbf{Y}|\mathbf{F}) \rangle_{q(\mathbf{F})} = \langle \varphi_0(r(\mathbf{Y}, \mathbf{F})) \rangle_{q(\mathbf{F})} \geq \varphi_0(\langle r(\mathbf{Y}, \mathbf{F}) \rangle_{q(\mathbf{F})}).$$

where we calculate the expectation of the quadratic form $r(\mathbf{Y}, \mathbf{F})$ as

$$\langle r(\mathbf{Y}, \mathbf{F}) \rangle_{q(\mathbf{F})} = r(\mathbf{Y}, \mathbf{K}_{NM}\mathbf{K}_{MM}^{-1}\mathbf{M}) + \beta D\mathrm{tr}(\mathbf{K}_{NN} - \mathbf{K}_{NM}\mathbf{K}_{MM}^{-1}\mathbf{K}_{MN})$$
$$+ \beta \sum_{d=1}^{D} \mathrm{tr}(\mathbf{K}_{NM}\mathbf{K}_{MM}^{-1}\boldsymbol{\Sigma}_d\mathbf{K}_{MM}^{-1}\mathbf{K}_{MN}).$$

Denote by $h(\mathbf{Y}, \mathbf{X}) = \langle\langle \log p(\mathbf{Y}|\mathbf{F}) \rangle_{q(\mathbf{F})}\rangle_{q(\mathbf{X})}$. Then by Jensen's inequality again we have

$$h(\mathbf{Y}, \mathbf{X}) \geq \varphi_0(\langle\langle r(\mathbf{Y}, \mathbf{F}) \rangle_{q(\mathbf{F})}\rangle_{q(\mathbf{X})}) =: h^*(\mathbf{Y}, \mathbf{X}).$$

Define $\psi_0 = \mathrm{tr}(\langle \mathbf{K}_{NN} \rangle_{q(\mathbf{X})})$, $\Psi_1 = \langle \mathbf{K}_{NM} \rangle_{q(\mathbf{X})}$, and $\Psi_2 = \langle \mathbf{K}_{MN}\mathbf{K}_{NM} \rangle_{q(\mathbf{X})}$. Further we calculate the expectations of quadratic terms similarly

$$\langle\langle r(\mathbf{Y}, \mathbf{F}) \rangle_{q(\mathbf{F})}\rangle_{q(\mathbf{X})} = \beta D[\psi_0 - \mathrm{tr}(\mathbf{K}_{MM}^{-1}\Psi_2)] + \beta \sum_{d=1}^{D} \mathrm{tr}(\mathbf{K}_{MM}^{-1}\boldsymbol{\Sigma}_d\mathbf{K}_{MM}^{-1}\Psi_2)$$
$$+ r(\mathbf{Y}, \Psi_1\mathbf{K}_{MM}^{-1}\mathbf{M}) + \beta\mathrm{tr}(\mathbf{M}^{\mathsf{T}}\mathbf{K}_{MM}^{-1}(\Psi_2 - \Psi_1^{\mathsf{T}}\Psi_1)\mathbf{K}_{MM}^{-1}\mathbf{M}).$$

We compute the lower bounds for two K-L divergence terms $\mathrm{KL}_{\mathbf{U}} := \mathrm{KL}(q(\mathbf{U})\|p(\mathbf{U}))$ and $\mathrm{KL}_{\mathbf{X}} := \mathrm{KL}(q(\mathbf{X})\|p(\mathbf{X}))$ by similar argument with Jensen's inequality and expectation of quadratic forms. Details are left to Appendix B.2 for interested readers.

### 3.2.2 SUMMARY OF ELBO

Denote by $\varphi(r; \Sigma, D) := -\frac{D}{2}\log|\Sigma| + \frac{ND}{2}\left(\frac{q}{2}-1\right)\log r - \frac{1}{2}r^{\frac{q}{2}}$. We summarize the final ELBO $\mathcal{L}^*(q(\mathbf{X}))$ as follows:

$$\log p(\mathbf{Y}) \geq \mathcal{L}(q(\mathbf{X})) = \int q(\mathbf{X})q(\mathbf{U})p(\mathbf{F}|\mathbf{U}, \mathbf{X}) \log \frac{p(\mathbf{Y}|\mathbf{F})p(\mathbf{U})p(\mathbf{X})}{q(\mathbf{U})q(\mathbf{X})} d\mathbf{F}d\mathbf{U}d\mathbf{X}$$
$$\geq \mathcal{L}^*(q(\mathbf{X})) := h^*(\mathbf{Y}, \mathbf{X}) - \mathrm{KL}_{\mathbf{U}}^* - \mathrm{KL}_{\mathbf{X}}^*,$$
$$h^*(\mathbf{Y}, \mathbf{X}) = \varphi(r_{\mathbf{Y}}; \beta^{-1}\mathbf{I}_N, D),$$
$$r_{\mathbf{Y}} = r(\mathbf{Y}, \Psi_1\mathbf{K}_{MM}^{-1}\mathbf{M}) + \beta\mathrm{tr}(\mathbf{M}^{\mathsf{T}}\mathbf{K}_{MM}^{-1}(\Psi_2 - \Psi_1^{\mathsf{T}}\Psi_1)\mathbf{K}_{MM}^{-1}\mathbf{M})$$
$$+ \beta D[\psi_0 - \mathrm{tr}(\mathbf{K}_{MM}^{-1}\Psi_2)] + \beta \sum_{d=1}^{D} \mathrm{tr}(\mathbf{K}_{MM}^{-1}\boldsymbol{\Sigma}_d\mathbf{K}_{MM}^{-1}\Psi_2), \quad (9)$$
$$-\mathrm{KL}_{\mathbf{U}}^* = \frac{1}{2}\sum_{d=1}^{D}\log|\boldsymbol{\Sigma}_d| + \varphi\left(\mathrm{tr}(\mathbf{M}^{\mathsf{T}}\mathbf{K}_{MM}^{-1}\mathbf{M}) + \sum_{d=1}^{D}\mathrm{tr}(\boldsymbol{\Sigma}_d\mathbf{K}_{MM}^{-1}); \mathbf{K}_{MM}, D\right),$$
$$-\mathrm{KL}_{\mathbf{X}}^* = \frac{1}{2}\sum_{n=1}^{N}\log|\mathbf{S}_n| + \varphi\left(\mathrm{tr}(\boldsymbol{\mu}^{\mathsf{T}}\boldsymbol{\mu}) + \sum_{n=1}^{N}\mathrm{tr}(\mathbf{S}_n); \mathbf{I}_N, Q\right),$$

where $\psi_0 = \mathrm{tr}(\langle \mathbf{K}_{NN} \rangle_{q(\mathbf{X})})$, $\Psi_1 = \langle \mathbf{K}_{NM} \rangle_{q(\mathbf{X})}$, and $\Psi_2 = \langle \mathbf{K}_{MN}\mathbf{K}_{NM} \rangle_{q(\mathbf{X})}$ (Appendix B.1).

**Remark 4.** *The variational solution $q(\mathbf{X})$ can be obtained by maximizing the ELBO (9) with respect to the variational parameters $(\boldsymbol{\mu}, \{\mathbf{S}_n\}, \tilde{\mathbf{X}}, \mathbf{M}, \{\boldsymbol{\Sigma}_d\})$ and kernel parameters $(\alpha, \beta, \boldsymbol{\gamma})$.*

**Remark 5.** *When $q = 2$, $\varphi(r; \Sigma, D) = -\frac{D}{2}\log|\Sigma| - \frac{1}{2}r$ with $r = r(\mathbf{Y}, \Psi_1\mathbf{K}_{MM}^{-1}\mathbf{M})$ becomes the log-density of matrix normal $\mathcal{MN}_{N\times D}(\Psi_1\mathbf{K}_{MM}^{-1}\mathbf{M}, \beta^{-1}\mathbf{I}_N, \mathbf{I}_D)$. Then the ELBO (9) reduces to Equation (7) of (SVGP Hensman et al., 2015) with an extra term $\beta\mathrm{tr}(\mathbf{M}^{\mathsf{T}}\mathbf{K}_{MM}^{-1}(\Psi_2 - \Psi_1^{\mathsf{T}}\Psi_1)\mathbf{K}_{MM}^{-1}\mathbf{M})$, not computed when $q = 2$ falling back to GP-LVM. The computational complexity, $\mathcal{O}(NM^2)$, remains the same as SVGP and GP-LVM (Titsias & Lawrence, 2010).*

### 3.3 WHAT $q$?

While the parameter $q > 0$ regularizes the learned latent representations (See Section 4), it remains as a question to automatically search for the optimal $q^*$, rather than manually pick one. Therefore, we impose a Gamma prior, $q \sim \Gamma(\alpha_0, \beta_0)$, and jointly optimize its posterior for the optimal $q^*$.

Note, the target function, ELBO (9), involves $q$ only through the function $\varphi$, which is concave in $q$ ($\frac{d^2\varphi}{dq^2} = -\frac{1}{2}r^{\frac{q}{2}}(\frac{1}{2}\log r)^2 \leq 0$). Therefore, if we block-update parameters in the overall optimization procedure, maximizing ELBO with respect to $q$ should return a unique solution, though the joint densities could be rather complicated (See Figure C.1). In the numerical experiments (Section 4), we adopt $\alpha_0 = 4$ and $\beta_0 = 2$ to make the prior mass concentrated in $(1, 2)$. Figure B.1 investigates $\varphi$ as a function of $q$ that plays a key role in finding the optimal $q^*$.

### 3.4 PREDICTION

With Bayesian QEP-LVM, we can make predictions for test data $\mathbf{y}_*$. As in Bayesian GP-LVM (Titsias & Lawrence, 2010), we can compute the predictive density $p(\mathbf{y}_*|\mathbf{Y})$ for some test data $\mathbf{y}_*$.

Given the latent variables $\mathbf{X}$ obtained for the training data $\mathbf{Y}$ and a new test latent variable $\mathbf{x}_*$, the predictive density can be written and approximated as follows

$$p(\mathbf{y}_*|\mathbf{Y}) = \frac{\int p(\mathbf{y}_*, \mathbf{Y}|\mathbf{X}, \mathbf{x}_*)p(\mathbf{X}, \mathbf{x}_*)d\mathbf{X}d\mathbf{x}_*}{\int p(\mathbf{Y}|\mathbf{X})p(\mathbf{X})d\mathbf{X}} \approx \exp\left\{\mathcal{L}(q(\mathbf{X}, \mathbf{x}_*)) - \mathcal{L}(q(\mathbf{X}))\right\} \quad (10)$$

where $\mathcal{L}$ is the ELBO as in (9). Specifically, we train a Bayesian QEP-LVM based on $\mathbf{Y}$ to get $\mathcal{L}(q(\mathbf{X}))$. In the testing stage, we append $\mathbf{y}_*$ to $\mathbf{Y}$ on which we train another Bayesian QEP-LVM with augmented variational distribution $q(\mathbf{X}, \mathbf{x}_*) \sim \text{q-ED}([\boldsymbol{\mu}, \boldsymbol{\mu}_*], \text{diag}(\{[\mathbf{S}_n, \mathbf{S}_*]\}))$ to obtain $\mathcal{L}(q(\mathbf{X}, \mathbf{x}_*))$.

The predictive density (10) can be utilized to build generative models to classify labels. Suppose in addition to data $\mathbf{Y}$, we have labels $\mathbf{t} = \{t_i\}_{i=1}^N$ falling in $K$ categories. Let $\mathbf{Y}^{(k)} = \{\mathbf{y}_i|t_i = k\}$. Then we train $K$ LVMs each based on one of $\{\mathbf{Y}^{(k)}\}_{k=1}^K$ and consider the following classifier:

$$\hat{t}_* = \arg\max_k p(\mathbf{y}_*|\mathbf{Y}^{(k)})p(t_* = k).$$

This will be further investigated in Section 4.2 and Section 4.3.

## 4 NUMERICAL EXPERIMENTS

In this section, we demonstrate the regularization effect of QEP-LVM on latent representation learning through the parameter $q > 0$ and compare it with GP-LVM ($q = 2$). We investigate the latent representations of multiple datasets learned by QEP-LVMs with changing $q > 0$ and observe the regularization effect that smaller $q$ tends to contract the latent space. Compared with GP-LVM as a QEP-LVM for fixed $q = 2$, optimizing the parameter $q$ often leads to a superior latent representation with enhanced interpretability. Throughout this section, we use the kernel (6) and the Gamma prior $\Gamma(4, 2)$ if varying $q > 0$ in the Bayesian framework. Since autoencoder is equivalent to PCA when all of its activation functions are linear, we also include the probabilistic version, variational autoencoder (VAE) (Kingma & Welling, 2014), for comparison. All the examples are implemented in `GPyTorch` (Gardner et al., 2018) and the computer codes are publicly available at `https://github.com/lanzithinking/Reg_Rep`.

### 4.1 SWISS ROLL

First we consider the Swiss roll dataset (Marsland, 2014) usually used in manifold learning. Illustrated in upper left panel of Figure 1, this dataset consists of a 3d point cloud resembling the food with the same name. Because the 1000 points mainly sit on a curved surface, we aim to learn a 2d latent subspace with GP-LVM and QEP-LVM respectively. As shown in the lower left panel of Figure 1, PCA returns a result seemingly by compressing the 3d cloud from above (along $-z$ axis). We then train VAE with 5 dense layers and the resulting 2d latent representation identified by the two largest variances of latent distribution resembles that of PCA. QEP-LVMs are trained for

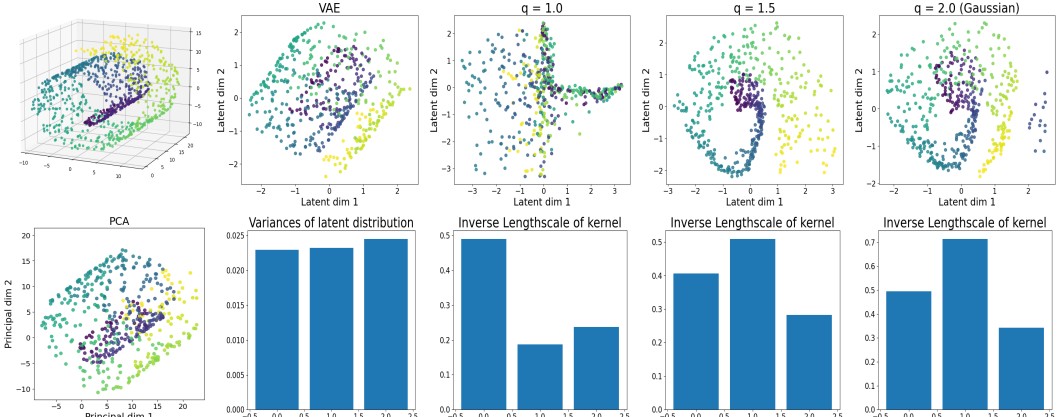

Figure 1: Latent representation of Swiss roll dataset. Upper left: 3d cloud of 1000 points; lower left: PCA in 2d principal space; right 4 columns: 2d latent representations (upper row) by VAE and QEP-LVMs with $q = 1.0, 1.5$ and $2.0$ (GP-LVM) showing a regularization effect via the parameter $q$, and the corresponding variances of latent distribution (VAE) and inverse length-scales $\boldsymbol{\gamma}$ ordered on the $x$-axes (lower row). Colors are used to aid visualization but not for training.

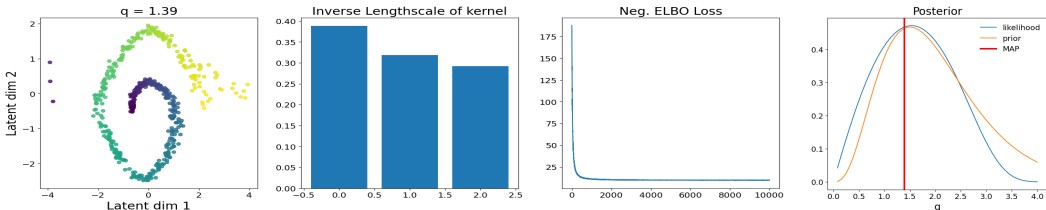

Figure 2: Latent representation of Swiss roll dataset output by QEP-LVM with optimal $q^* = 1.39$ (red line in the rightmost panel) found in the Bayesian framework.

different $q$s with 25 inducing points. The upper row of right three panels in Figure 1 compares the latent representations output by QEP-LVMs with $q = 1.0, 1.5$ and $2.0$ (GP-LVM) which all appear as projections from an outward view (along $x$ axis). As $q$ decreases from 2.0 to 1.0, the learned latent representations contract towards axes, verifying its regularization effect. The latent result with $q = 1.5$ is the best representing the latent "roll" structure among these plots. Most of the inverse length-scales ($\boldsymbol{\gamma}$) of the kernel in the lower row imply 2 dominant dimensions (used to plot 2d latent representations), indicating an intrinsic dimensionality for this dataset.

Next, we let $q > 0$ be a random variable imposed with a Gamma prior and jointly optimize $q$ with other parameters in the QEP-LVM. Figure C.1 plots the pairwise densities of $q$ and length-scale $l = 1/\gamma$, which highlights the difficulty of joint optimization (different scales, complex landscape, etc). As shown in the rightmost panel of Figure 2, the optimal regularization parameter is attained at $q^* = 1.39$. The resulting latent representation, as in the leftmost panel of Figure 2, reflects a clear latent "roll" structure.

We dig a hole in the 3d point cloud along the curved surface to make a "Swiss hole" dataset in Appendix C.1. We also include latent representations by Isomap and t-SNE in Figure C.2 for comparison. Then we repeat the same experiments for such dataset and present similar results in Figure C.3 and Figure C.4. Again QEP-LVM with regularization parameter around 1.5 outputs latent representation having a "roll" with a "hole" structure better than GP-LVM and VAE.

## 4.2 OIL FLOW

Next, we demonstrate the behavior of QEP-LVM and contrast it with GP-LVM using the canonical multi-phase oil-flow dataset (Bishop & James, 1993) that consists of 1000 observations (12-

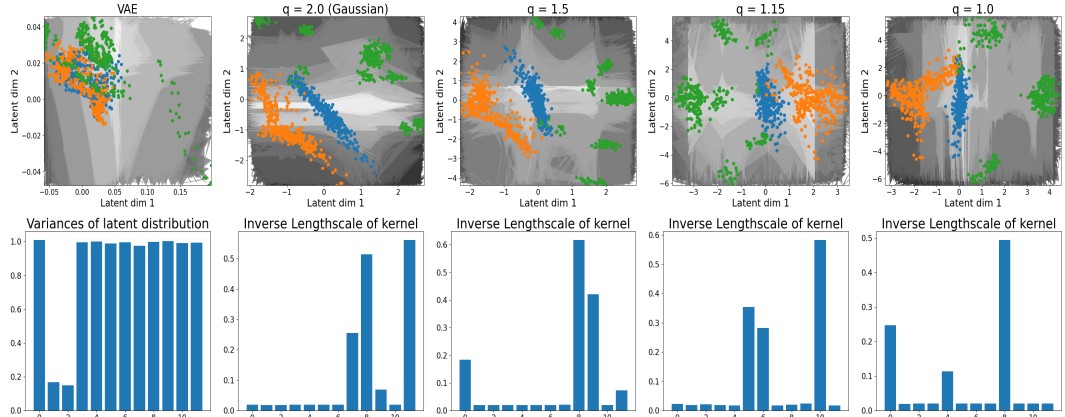

Figure 3: Latent representation of oil flow dataset. Upper: 2d latent representations by VAE and QEP-LVMs with $q = 2.0$ (Gaussian), $1.5, 1.15$ and $1.0$ showing a regularization effect via the parameter $q$. Lower: the corresponding variances of latent distribution and inverse length-scales $\gamma$. Colors for points label three classes but are not used for training. Gray-scale colors indicate uncertainty approximated by the variational distribution.

dimensional) belonging to three known classes corresponding to different phases of oil-flow. This dataset is also used in Lawrence (2003); Titsias & Lawrence (2010). We train the models only on the 12 covariates (no labels used) with the same settings as the above example in Section 4.1. The upper row of Figure 3 visualizes 2d slices of the latent spaces by VAE (identified by the largest two latent variances) and by QEP-LVMs corresponding to the most dominant latent dimensions (identified by the largest two inverse length-scales $\gamma$) with $q = 2.0$ (Gaussian), $1.5, 1.15$ and $1.0$ respectively. The gray-scale color indicates the uncertainty (computed from $\mathbf{S}_n$ in $q(\mathbf{X})$). Again we observe the regularization effect by the parameter $q > 0$ of QEP-LVM in latent representation learning: smaller $q$ leads to more regularization on the learned representations and hence yields more aggregated clusters, as illustrated by the green class. In the lower row of Figure 3, except VAE, all the QEP-LVM models unanimously identify 3 dominant latent dimensions despite their orders.

We also vary $q > 0$ with the same Gamma prior and obtain the optimal $q^* = 1.15$. The results are shown in Figure C.6. With each trained QEP-LVM, we utilize the predictive density to build a generative classifier as described in Section 3.4. Then we compare these classifiers on a common test dataset and repeat the experiments for 10 different random seeds. Table 1 summarizes the performance in terms of accuracy (ACC), area under ROC curve (AUC), adjusted rand index (ARI) score, normalized mutual information (NMI) score, log predictive probability (LPP) values, and running time (per class). The models with $q = 1.0$ and $1.15$ are generally better than the other two with $q = 1.5$ and $2.0$ (Gaussian) probably because the former two have labeled points more separated from each class in the latent subspace. The best classifier (ACC/AUC) coincides with the one for optimal $q^* = 1.15$. Note that LPPs are not comparable with each other since they are from different models with different likelihoods. Though not used in training, the class labels can be identified as cluster assignments. Then QEP-LVM with optimal $q^* = 1.15$ also attains the highest ARI and NMI scores which are metrics for evaluating clustering algorithms.

Table 1: Classification on oil flow data based on learned latent representation: accuracy (ACC), area under ROC curve (AUC), adjusted rand index (ARI) score, normalized mutual information (NMI) score, log predictive probability (LPP) values, and running time (per class) by various Bayesian QEP-LVMs. Result in each cell are averaged over 10 experiments with different random seeds; values after $\pm$ are standard errors of these repeated experiments.

| Model ($q$) | ACC | AUC | ARI | NMI | LPP | time/class |
|---|---|---|---|---|---|---|
| 1.0 | $0.70 \pm 0.04$ | $0.86 \pm 0.03$ | $0.35 \pm 0.05$ | $0.36 \pm 0.05$ | $-10.42 \pm 0.42$ | $211.92 \pm 12.53$ |
| 1.15 | $\mathbf{0.73} \pm 0.04$ | $\mathbf{0.89} \pm 0.02$ | $\mathbf{0.38} \pm 0.05$ | $\mathbf{0.39} \pm 0.05$ | $-9.21 \pm 0.25$ | $212.30 \pm 13.17$ |
| 1.5 | $0.67 \pm 0.05$ | $0.83 \pm 0.03$ | $0.30 \pm 0.07$ | $0.33 \pm 0.06$ | $\mathbf{-7.70} \pm 0.25$ | $206.91 \pm 13.58$ |
| 2.0(Gaussian) | $0.68 \pm 0.05$ | $0.83 \pm 0.03$ | $0.34 \pm 0.07$ | $0.37 \pm 0.07$ | $-8.78 \pm 0.19$ | $198.47 \pm 14.42$ |

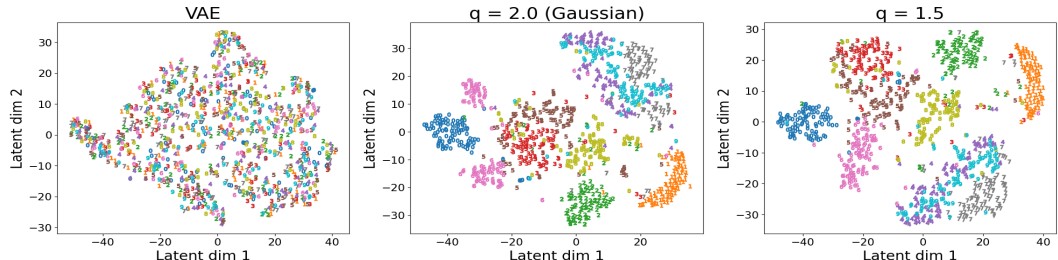

Figure 4: Latent representations of MNIST database by VAE (left), GP-LVM (middle), QEP-LVM with $q = 1.5$ (right). For the convenience of visualization, 10-dimensional latent spaces learned by these algorithms are projected to 2-d subspace by t-SNE respectively.

### 4.3 MNIST

Lastly, we consider the MNIST database (Lecun et al., 1998) consisting of $60,000$ training and $10,000$ testing handwritten digits of size $28 \times 28$. We suppress the labels and use these images alone to train QEP-LVMs for $q = 1.5$ and $2.0$ (Gaussian) with 128 inducing points. We set the latent dimension to be 10 and use t-SNE to project latent spaces by VAE, GP-LVM and QEP-LVM ($q = 1.5$) to 2d subspaces in Figure 4 for better visualization. Unlike VAE generating unstructured latent representation with all digits mixed up, outputs by LVMs are much more interpretable. While GP-LVM has segregated clusters of digits 6 and 7 respectively, QEP-LVM has more concentrated clusters with each further apart from others, despite confusion-prone groups 3-5-8 and 4-9-(7). Figure C.7 compares the pairwise (10 digits are divided into 5 groups of confusing pairs) latent representations by GP-LVM (lower) and QEP-LVM ($q = 1.5$, upper). QEP-LVM ($q = 1.5$) has latent representations better separated in the pairs of $(0, 6)$, $(2, 3)$ and $(5, 8)$ compared with GP-LVM. Seen from Figure C.8, QEP-LVM ($q = 1.5$) generates sample digits of 3, 5 clearer than GP-LVM. Table 2 compares the generative classifiers constructed as in Section 3.4. QEP-LVM ($q = 1.5$) outperforms GP-LVM ($q = 2.0$) in terms of ACC and AUC. If we view the class labels (not used in training) as cluster assignments, QEP-LVM ($q = 1.5$) also has better ARI and NMI scores for clustering.

Table 2: Classification on handwritten digits (MNIST) based on learned latent representation: accuracy (ACC), area under ROC curve (AUC), adjusted rand index (ARI) score, normalized mutual information (NMI) score, log predictive probability (LPP) values, and running time (per class) by various Bayesian QEP-LVMs. Results in each cell are averaged over 10 experiments with different random seeds; values after $\pm$ are standard errors of these repeated experiments.

| Model ($q$) | ACC | AUC | ARI | NMI | LPP | time/class |
|---|---|---|---|---|---|---|
| 1.5 | **0.973** $\pm$ 0.012 | **0.986** $\pm$ 0.0065 | **0.943** $\pm$ 0.024 | **0.952** $\pm$ 0.020 | **-26453.5** $\pm$ 481.9 | 129.91 $\pm$ 4.60 |
| 2.0 (Gaussian) | 0.965 $\pm$ 0.012 | 0.981 $\pm$ 0.0062 | 0.923 $\pm$ 0.025 | 0.938 $\pm$ 0.019 | -279414.9 $\pm$ 5184.6 | 127.18 $\pm$ 0.29 |

## 5  CONCLUSION

In this paper, we introduce a novel Bayesian latent variable model based on the recently proposed Q-exponential process (Li et al., 2023) (QEP-LVM) for latent representation learning. Q-EP empowers the LVM with flexible regularization that controls the complexity of the learned latent representations often with improved interpretability compared with GP-LVM, a special case of QEP-LVM for $q = 2$. The theoretic connection between QEP-LVM and probabilistic PCA has been established. Using three examples, we demonstrate the advantage of the proposed methodology in terms of quality of learned latent representation and quantitative performance on the derived generative models.

It remains unknown how to efficiently deal with missing data, one of the possible defects of the proposed QEP-LVM. Data imputation based on the variational distribution for latent function, $q(\mathbf{f}_* | \mathbf{X})$, has no tractable form when integrating with respect to $q(\mathbf{X})$ as for GP-LVM (Equation (21) in Titsias & Lawrence (2010)). Perhaps Monte Carlo approximation is unavoidable. Another interesting direction is to generalize QEP-LVM to deep Q-EP as deep GP (Damianou & Lawrence, 2013), which we will treat in a separate paper.

ACKNOWLEDGMENTS

SL is supported by NSF grant DMS-2134256.

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

# Supplement Document for "Bayesian Regularization of Latent Representation"

## A  PROOFS

*Proof of Theorem 3.1.* The gradients of log-likelihood (5) with respect to $\mathbf{K}$ can be found as

$$\frac{\partial L}{\partial \mathbf{K}} = -\frac{D}{2}\mathbf{K}^{-1} - \left[\frac{ND}{2}\left(\frac{q}{2} - 1\right)r^{-1} - \frac{1}{2}\frac{q}{2}r^{\frac{q}{2}-1}\right]\mathbf{K}^{-1}\mathbf{Y}\mathbf{Y}^{\mathsf{T}}\mathbf{K}^{-1}. \tag{11}$$

The MLE for $\mathbf{X}$ can be found by setting $\frac{\partial L}{\partial \mathbf{X}} = 2\alpha^{-1}\frac{\partial L}{\partial \mathbf{K}}\mathbf{X} = 0$, which leads to

$$\mathbf{X} = \left[-ND\left(\frac{q}{2} - 1\right)r^{-1} + \frac{q}{2}r^{\frac{q}{2}-1}\right]D^{-1}\mathbf{Y}\mathbf{Y}^{\mathsf{T}}\mathbf{K}^{-1}\mathbf{X}. \tag{12}$$

Now suppose we have the formal solution for (12) as $\mathbf{X} = \mathbf{U}\mathbf{L}\mathbf{V}$, where $\mathbf{L}$ is an $N \times Q$ matrix whose only nonzero entries $\{l_i\}$ are on the main diagonal to be determined. Based on the formal solution of $\mathbf{X}$, we have

$$\mathbf{K} = \mathbf{U}(\alpha^{-1}\mathbf{L}\mathbf{L}^{\mathsf{T}} + \beta^{-1}\mathbf{I})\mathbf{U}^{\mathsf{T}}, \quad r(\mathbf{Y}) = \mathrm{tr}(\mathbf{K}^{-1}\mathbf{Y}\mathbf{Y}^{\mathsf{T}}) = \sum_{i=1}^{D \wedge Q} D\lambda_i(\alpha^{-1}l_i^2 + \beta^{-1})^{-1}.$$

Denote by $h(r) := -ND\left(\frac{q}{2} - 1\right)r^{-1} + \frac{q}{2}r^{\frac{q}{2}-1}$. We substitute the above quantities into (12) and get

$$\mathbf{U}\mathbf{L}\mathbf{V} = \mathbf{U}\mathbf{\Lambda}(\alpha^{-1}\mathbf{L}\mathbf{L}^{\mathsf{T}} + \beta^{-1}\mathbf{I})^{-1}\mathbf{L}\mathbf{V}h(r),$$

which reduces to

$$l_i = \lambda_i(\alpha^{-1}l_i^2 + \beta^{-1})^{-1}l_i h(r), \quad i = 1, \cdots, D \wedge Q.$$

Let $h(r) = c$ with $c$ to be determined. Assume $l_i \neq 0$. Then we can solve $l_i = \sqrt{\alpha(c\lambda_i - \beta^{-1})}$. This yields $r = c^{-1}D(D \wedge Q)$. Hence

$$h(r) = -N\left(\frac{q}{2} - 1\right)c(D \wedge Q)^{-1} + \frac{q}{2}[D(D \wedge Q)]^{\frac{q}{2}-1}c^{1-\frac{q}{2}} = c.$$

And it solves $c = D^{1-\frac{2}{q}}(D \wedge Q)\left[\frac{q}{2(D \wedge Q) + (q-2)N}\right]^{\frac{2}{q}}$. Hence the proof is completed.

$\square$

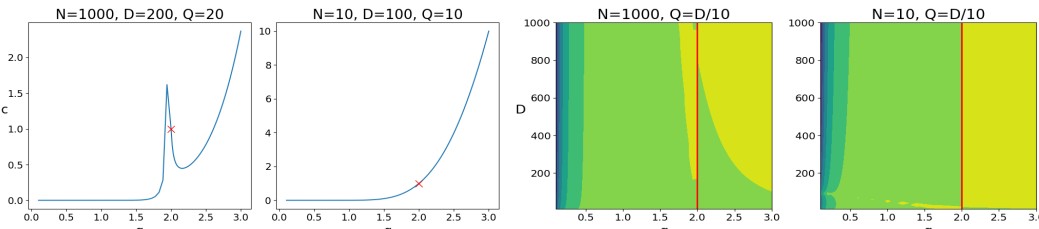

Figure A.1: Function $c(q)$ in Theorem 3.1 that regularizes the singular values of latent variable $\mathbf{X}$. Left two panels: $c(q)$ for fixed $N, D, Q$; right two panels: $c(q)$ for fixed $N$ and $Q = D/10$.

## B  VARIATIONAL BAYES FOR QEP-LVM

### B.1  EXPECTED KERNEL TERMS

We compute $\psi_0 = \mathrm{tr}(\langle \mathbf{K}_{NN} \rangle_{q(\mathbf{X})})$, $\Psi_1 = \langle \mathbf{K}_{NM} \rangle_{q(\mathbf{X})}$, and $\Psi_2 = \langle \mathbf{K}_{MN} \mathbf{K}_{NM} \rangle_{q(\mathbf{X})}$ as follows:

$$\psi_0^n = \int k(\mathbf{x}_n, \mathbf{x}_n) \text{q-ED}(\mathbf{x}_n | \boldsymbol{\mu}_n, \mathbf{S}_n) d\mathbf{x}_n,$$

$$(\Psi_1)_{nm} = \int k(\mathbf{x}_n, \mathbf{z}_m) \text{q-ED}(\mathbf{x}_n | \boldsymbol{\mu}_n, \mathbf{S}_n) d\mathbf{x}_n,$$

$$(\Psi_2^n)_{mm'} = \int k(\mathbf{x}_n, \mathbf{z}_m) k(\mathbf{z}_{m'}, \mathbf{x}_n) \text{q-ED}(\mathbf{x}_n | \boldsymbol{\mu}_n, \mathbf{S}_n) d\mathbf{x}_n.$$

With ARD SE kernel (6), we have $\psi_0 = N\alpha^{-1}$. While the integration in $\Psi_1$ and $\Psi_2$ is intractable in general, we can compute them using Monte Carlo approximation. Alternatively, we approximate

$$(\Psi_1)_{nm} \approx \alpha^{-1} \exp\left\{ -\frac{1}{2} \langle (\mathbf{x}_n - \mathbf{z}_m)^{\mathsf{T}} \mathrm{diag}(\boldsymbol{\gamma})(\mathbf{x}_n - \mathbf{z}_m) \rangle_{q(\mathbf{x}_n)} \right\}$$

$$= \alpha^{-1} \exp\left\{ -\frac{1}{2} [(\boldsymbol{\mu}_n - \mathbf{z}_m)^{\mathsf{T}} \mathrm{diag}(\boldsymbol{\gamma})(\boldsymbol{\mu}_n - \mathbf{z}_m) + \mathrm{tr}(\mathrm{diag}(\boldsymbol{\gamma})\mathbf{S}_n)] \right\},$$

$$(\Psi_2^n)_{mm'} \approx \alpha^{-2} \exp\left\{ -\frac{1}{2} \sum_{\tilde{m}=m,m'} (\boldsymbol{\mu}_n - \mathbf{z}_{\tilde{m}})^{\mathsf{T}} \mathrm{diag}(\boldsymbol{\gamma})(\boldsymbol{\mu}_n - \mathbf{z}_{\tilde{m}})) + \mathrm{tr}(\mathrm{diag}(\boldsymbol{\gamma})\mathbf{S}_n) \right\}.$$

If we use the ARD linear form, $k(\mathbf{x}, \mathbf{x}') = \mathbf{x}^{\mathsf{T}} \mathrm{diag}(\boldsymbol{\gamma})\mathbf{x}'$, then we have

$$\psi_0^n = \mathrm{tr}(\mathrm{diag}(\boldsymbol{\gamma})(\boldsymbol{\mu}_n \boldsymbol{\mu}_n^{\mathsf{T}} + \mathbf{S}_n)), \quad (\Psi_1)_{nm} = \boldsymbol{\mu}_n^{\mathsf{T}} \mathrm{diag}(\boldsymbol{\gamma}) \mathbf{z}_m,$$

$$(\Psi_2^n)_{mm'} = \mathbf{z}_m^{\mathsf{T}} \mathrm{diag}(\boldsymbol{\gamma})(\boldsymbol{\mu}_n \boldsymbol{\mu}_n^{\mathsf{T}} + \mathbf{S}_n) \mathrm{diag}(\boldsymbol{\gamma}) \mathbf{z}_{m'}.$$

### B.2  LOWER BOUND FOR THE K-L DIVERGENCE ADDED TERMS

We also need to compute the K-L divergence $\mathrm{KL}_{\mathbf{U}} := \mathrm{KL}(q(\mathbf{U}) \| p(\mathbf{U}))$:

$$\mathrm{KL}_{\mathbf{U}} = \int q(\mathbf{U}) \log q(\mathbf{U}) d\mathbf{U} - \int q(\mathbf{U}) \log p(\mathbf{U}) d\mathbf{U} = -\mathcal{H}_q(\mathbf{U}) - \langle \log p(\mathbf{U}) \rangle_{q(\mathbf{U})}.$$

Denote by $r = \mathrm{vec}^{\mathsf{T}}(\mathbf{U} - \mathbf{M})^{\mathsf{T}} \mathrm{diag}(\{\boldsymbol{\Sigma}_d\})^{-1} \mathrm{vec}^{\mathsf{T}}(\mathbf{U} - \mathbf{M})$. Then $\log q(\mathbf{U}) = -\frac{1}{2} \sum_{d=1}^{D} \log |\boldsymbol{\Sigma}_d| + \frac{MD}{2}\left(\frac{q}{2} - 1\right) \log r - \frac{1}{2} r^{\frac{q}{2}}$. From (Proposition A.1. of Li et al., 2023) we know that $r^{\frac{q}{2}} \sim \chi^2(MD)$. Therefore

$$\mathcal{H}_q(\mathbf{U}) = \frac{1}{2} \sum_{d=1}^{D} \log |\boldsymbol{\Sigma}_d| + \frac{MD}{2}\left(\frac{q}{2} - 1\right) \frac{2}{q} \mathcal{H}(\chi^2(MD)) + \frac{MD}{2}$$

$$= \frac{1}{2} \sum_{d=1}^{D} \log |\boldsymbol{\Sigma}_d| + \frac{MD}{2}\left(1 - \frac{2}{q}\right) \left[\frac{MD}{2} + \log\left(2\Gamma\left(\frac{MD}{2}\right)\right) + \left(1 - \frac{MD}{2}\right) \psi\left(\frac{MD}{2}\right)\right] + \frac{MD}{2}.$$

Denote by $\varphi_1(r) := -\frac{D}{2} \log |\mathbf{K}_{MM}| + \frac{MD}{2}\left(\frac{q}{2} - 1\right) \log r - \frac{1}{2} r^{\frac{q}{2}}$. Then by Jensen's inequality

$$\langle \log p(\mathbf{U}) \rangle_{q(\mathbf{U})} = \langle \varphi_1(\mathrm{tr}(\mathbf{U}^{\mathsf{T}} \mathbf{K}_{MM}^{-1} \mathbf{U})) \rangle_{q(\mathbf{U})} \geq \varphi_1(\langle \mathrm{tr}(\mathbf{U}^{\mathsf{T}} \mathbf{K}_{MM}^{-1} \mathbf{U}) \rangle_{q(\mathbf{U})}),$$

$$\langle \mathrm{tr}(\mathbf{U}^{\mathsf{T}} \mathbf{K}_{MM}^{-1} \mathbf{U}) \rangle_{q(\mathbf{U})} = \mathrm{tr}(\mathbf{M}^{\mathsf{T}} \mathbf{K}_{MM}^{-1} \mathbf{M}) + \sum_{d=1}^{D} \mathrm{tr}(\boldsymbol{\Sigma}_d \mathbf{K}_{MM}^{-1}).$$

Therefore we have the K-L divergence added term $\mathrm{KL}_{\mathbf{U}}$ bounded by

$$-\mathrm{KL}_{\mathbf{U}} \geq -\mathrm{KL}_{\mathbf{U}}^* := \frac{1}{2} \sum_{d=1}^{D} \log |\boldsymbol{\Sigma}_d| + \varphi_0(\langle \mathrm{tr}(\mathbf{U}^{\mathsf{T}} \mathbf{K}_{MM}^{-1} \mathbf{U}) \rangle_{q(\mathbf{U})}).$$

Lastly, we bound the K-L divergence term $\mathrm{KL}_{\mathbf{X}} := \mathrm{KL}(q(\mathbf{X})\|p(\mathbf{X}))$ by similar argument:

$$-\mathrm{KL}_{\mathbf{X}} \geq -\mathrm{KL}_{\mathbf{X}}^* := \frac{1}{2} \sum_{n=1}^{N} \log |\mathbf{S}_n| + \varphi_2(\langle \mathrm{tr}(\mathbf{X}^{\mathsf{T}}\mathbf{X}) \rangle_{q(\mathbf{X})}),$$

where $\varphi_2(r) := \frac{NQ}{2}\left(\frac{q}{2}-1\right)\log r - \frac{1}{2}r^{\frac{q}{2}}$ and $\langle \mathrm{tr}(\mathbf{X}^{\mathsf{T}}\mathbf{X}) \rangle_{q(\mathbf{X})} = \mathrm{tr}(\boldsymbol{\mu}^{\mathsf{T}}\boldsymbol{\mu}) + \sum_{n=1}^{N}\mathrm{tr}(\mathbf{S}_n)$.

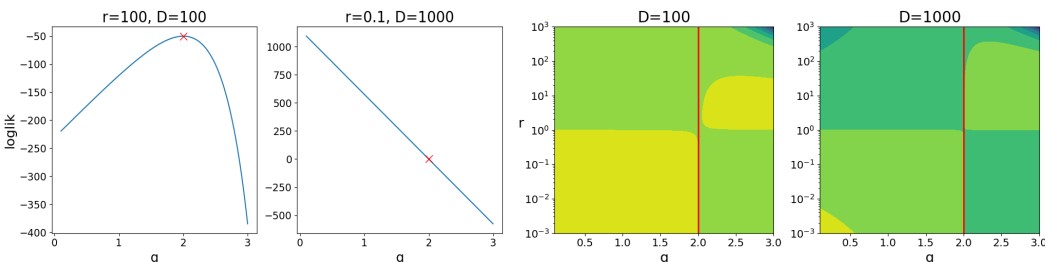

Figure B.1: Log-likelihood function $\varphi(q)$. Left two panels: $\varphi(q)$ for fixed $r, D$; right two panels: $\varphi(q)$ for fixed $D$.

# C   MORE NUMERICAL RESULTS

## C.1   SWISS HOLE

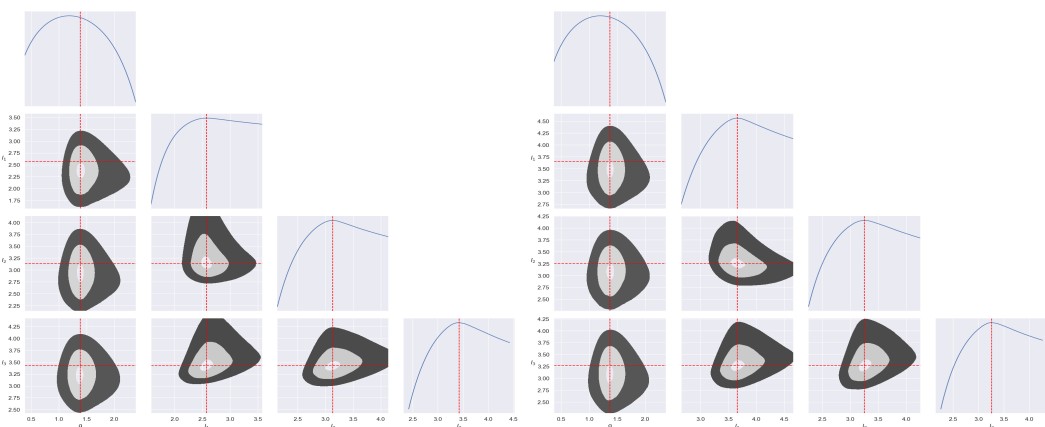

Figure C.1: Pairwise densities between regularization parameter $q$ and kernel length-scales $\mathbf{l} = (l_1, l_2, l_3)$ in Swiss roll (left) and Swiss hole (right) problems.

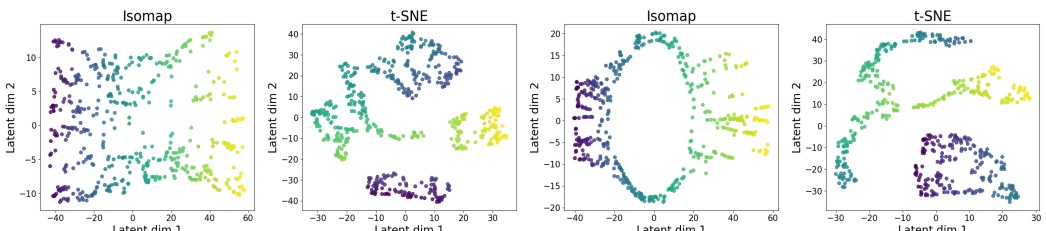

Figure C.2: Latent representation of Swiss roll (left two) and Swiss hole (right two) by Isomap and t-SNE respectively.

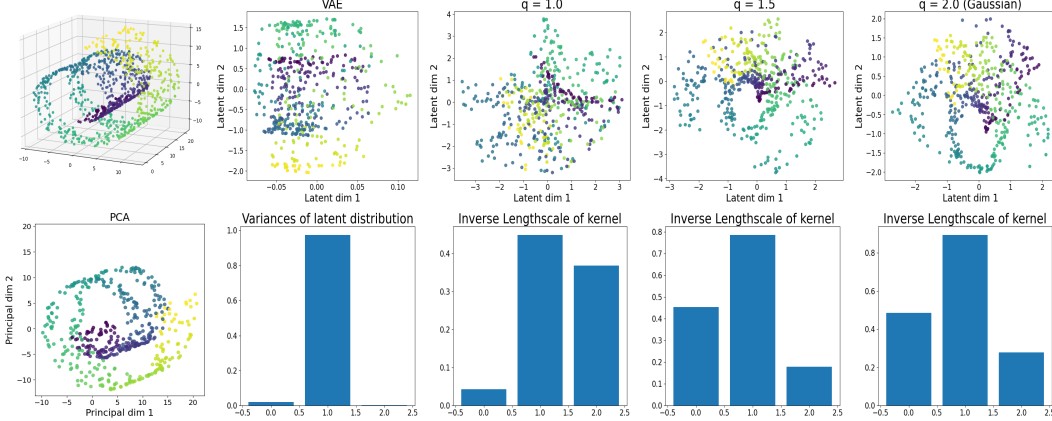

Figure C.3: Latent representation of Swiss hole dataset. Upper left: 3d cloud of 1000 points; lower left: PCA in 2d principal space; right 3 columns: 2d latent representations (upper row) by VAE and QEP-LVMs with $q = 1.0, 1.5$ and $2.0$ (GP-LVM) showing a regularization effect via the parameter $q$, and the corresponding variances of latent distribution (VAE) and inverse length-scales $\gamma$ (lower row). Colors are used to aid visualization but not for training.

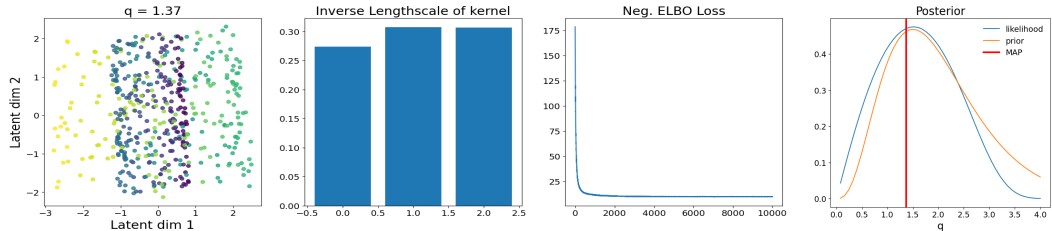

Figure C.4: Latent representation of Swiss hole dataset output by QEP-LVM with optimal $q^* = 1.37$ (red line in the rightmost panel) found in the Bayesian framework.

## C.2 ROBOT LEARNING

In this section, we consider the robot learning problem in Pahič et al. (2021). Robots can improve their performance by repeating desired behavior and updating the learned skill representation. So it is important to obtain the latent space of motor skills on which statistical and reinforcement learning can be applied to optimize robot's behavior. We focus on the task of robotic ball throwing at a target, which is realized by a 7 degree of freedom (DOF) arm Mitsubishi PA-10. Three DOFs of the robot, which contribute to its motion in the saggital plane, are used for the ball throwing. Dynamic movement primitives (DMPs) are used to represent smooth robot motion (Ijspeert et al., 2013) and with $N = 20$ DMP weights for each DOF we get 60 weights in the DMP parameter space. Here we test PCA, VAE, beta-VAE, and QEP-LVMs in learning the latent representation of motion skills.

In this dataset, there are no labels. We compare the latent spaces generated by different algorithms in Figure C.5. We set latent dimension to 10 and visualize the 2d subspaces projected by t-SNE algorithm. VAE and QEP-LVM ($q = 1.0$) output more structured latent spaces than others. In contrast, the latent representations by beta-VAE and GP-LVM appear more dispersed and unstructured. Among all the algorithms, only beta-VAE and QEP-LVM ($q = 1.0$) correctly identify the intrinsic dimensionality 3 (DOFs) of the latent space.

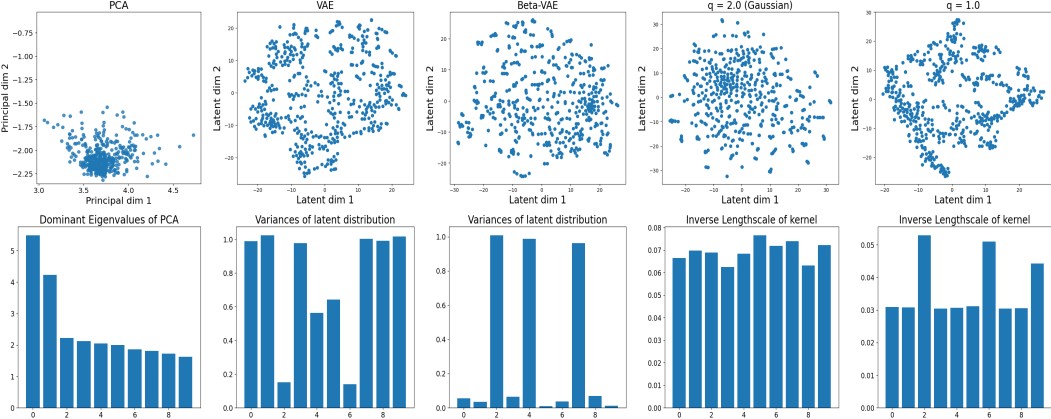

Figure C.5: Latent representation of robot learning dataset. Upper: 2d latent representations output by PCA, VAE, beta-VAE, GP-LVM and QEP-LVM ($q = 1.0$). Lower: latent dimensions indicated by dominant eigenvalues (PCA), variances of latent distribution (VAEs), and inverse length-scales $\gamma$ in QEP-LVMs. For the convenience of visualization, 10-dimensional latent spaces learned by these algorithms are projected to 2-d subspace by t-SNE respectively.

## C.3 OIL FLOW

## C.4 MNIST

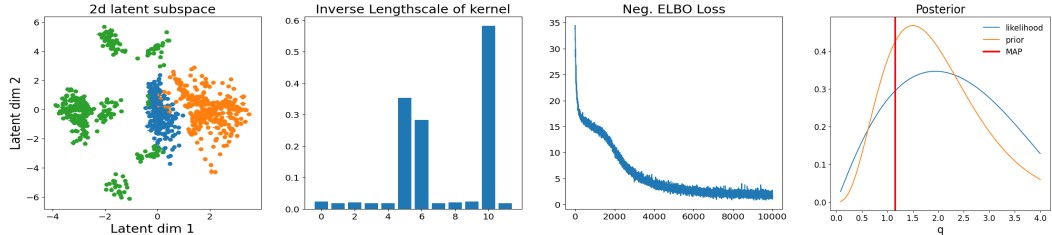

Figure C.6: Latent representation of oil flow dataset output by QEP-LVM with optimal $q^* = 1.57$ (red line in the rightmost panel) found in the Bayesian framework.

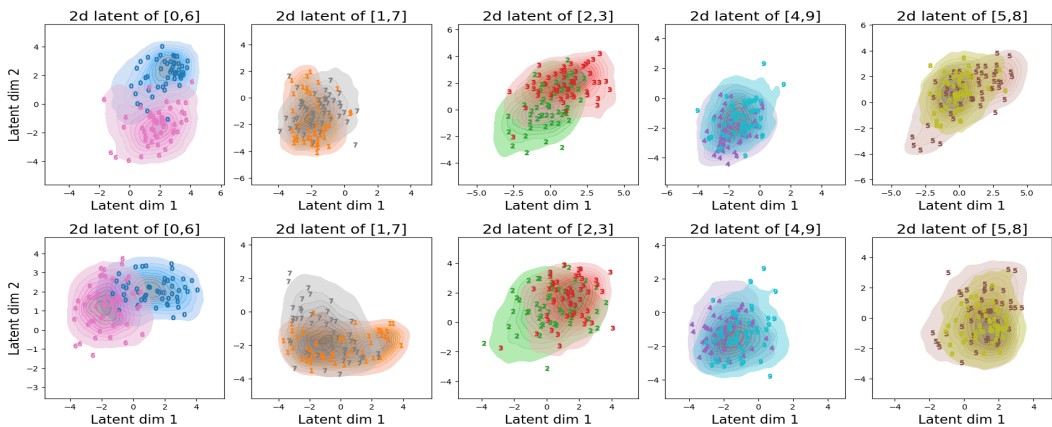

Figure C.7: Pairwise latent representations of MNIST database by QEP-LVMs with $q = 1.5$ (upper) and $q = 2.0$ (Gaussian, lower).

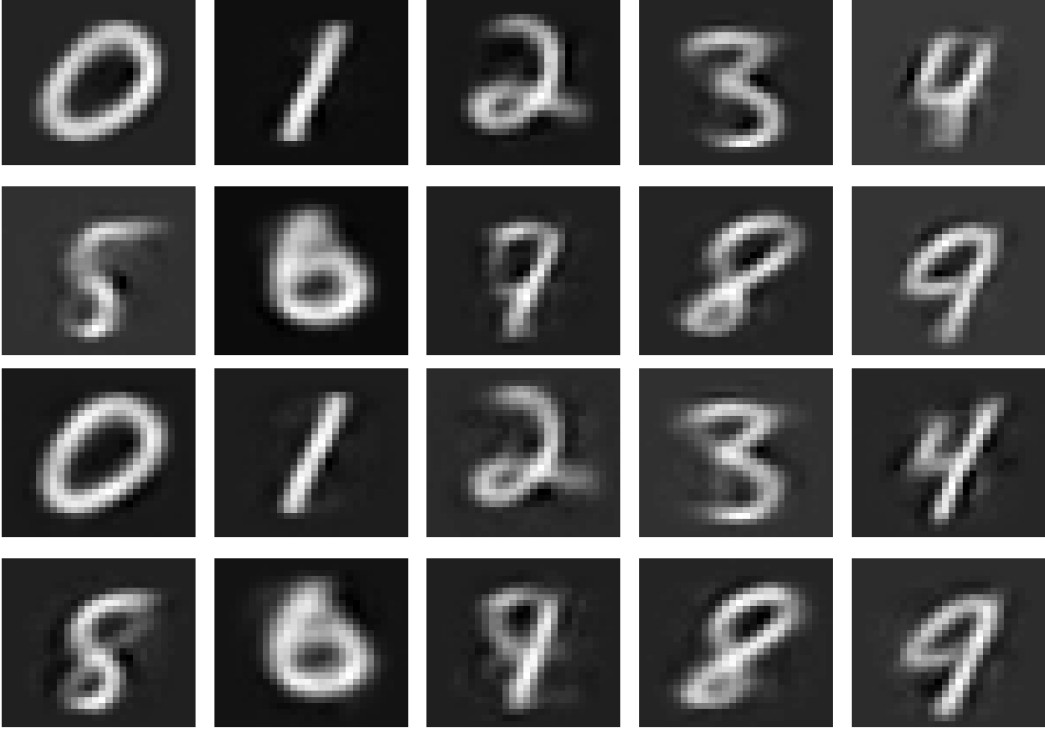

Figure C.8: Sample MNIST digits by QEP-LVMs with $q = 1.5$ (upper two rows) and $q = 2.0$ (Gaussian, lower two rows).

