# OpenReview forum: "Bayesian Regularization of Latent Representation"
_ICLR.cc/2025/Conference — ICLR 2025 Poster_

### Official Review · Reviewer_th3c · 2024-10-19

**Soundness:** 3
**Presentation:** 4
**Contribution:** 2
**Rating:** 6
**Confidence:** 4

**Summary:**

This paper extends the GP-LVM to a more general form, known as QEP-LVM, which regularizes the model using an Lq penalty. It establishes the equivalence between GP-LVM and a specific case of QEP-LVM. The experimental results conducted on three real-world datasets demonstrate the good performance of the proposed approach.

**Strengths:**

The paper is well-organized and easy to read.

**Weaknesses:**

1. It would be beneficial to include the time complexity analysis and a comparison of the running times between GP-LVM and QEP-LVM, especially for a relatively large data set -- MNIST.
2. For me, it represents a natural combination of QEP and LVM. While the creativity is incremental, I would like to see more applications in other areas where Gaussian processes are typically utilized.

**Questions:**

1. What is the practical meaning of q? Also, the author mentioned that it's based on the Lq penalty, but I'd like more explanation, like why q ranges from 1 to 2.
2. Does the choice of kernel impact the performance of the QEP-LVM?

---

> ### Author Response · Authors · 2024-11-20
> **added computation complexity and running times, clarified Lq penalty**
>
> Thank you for acknowledging the strengths of our paper. As commented in previous responses, the novelty of the proposed QEP-LVM lies in the flexibility on regularizing latent representations in various datasets. While the motivation is to generalize, rather than replacing GP-LVM, QEP-LVM, when successfully implemented, has the advantages including i) flexibility to regularize on the learned latent representations; ii) effective identification of the intrinsic latent dimensionality; iii) learning latent representations with better interpretability and adaptability for supervised/semi-supervised learning. They have all been well illustrated in numerical experiments.
>
> Weaknesses:
> 1. Yes, the complexity, $\mathcal O(NM^2)$ ($N$ the data size and $M$ the size of inducing points), has been added to Remark 5 (previous remark 4). Running times have also been added to both Tables.
> 2. As commented above, the creativity is not incremental. Claimed advantages have been well demonstrated by simulation and real data at various scales (from 3 dimensions to 784 dimensions). We have also added a robot learning example in Appendix C.2. All the examples support that the proposed QEP-LVM has potential impact on latent representation learning.
>
> Questions:
> 1. The practical meaning of $q$ is the degree of $L_q$ regularization. Remark 1 has been added to clarify the connection to the commonly known regularization based on $q$-vector norm. In theory $q>0$ but interesting applications usually take $q\in (0,2]$ in practice.
> 2. No, the performance of QEP-LVM is robust to the choice of kernels.
>
> All the relevant changes in the [**Rebuttal Revision**](/pdf/93b34ea5f98a2830c1c4afc11cc761fafe2430e5.pdf) are marked in $\textcolor{purple}{\text{purple}}$.

---

> > ### Comment · Reviewer_th3c · 2024-11-25
> >
> > Thank the authors for addressing most of my concerns and questions.
> >
> > My only remaining question is how do you conclude that "the performance of QEP-LVM is robust to the choice of kernels"?

---

> > > ### Author Response · Authors · 2024-11-26
> > >
> > > We did try other kernels. They may produce slightly different results. But the numerical conclusion (regularization effect by $q$, better interpretability of QEP-LVM) remains unchanged. We have revised our wording in the  [Rebuttal Revision](/pdf/dff2db890d6680986794b21ea627f3431b4f5a3e.pdf) with relevant changes marked in $\textcolor{purple}{purple}$.

---

> > > > ### Comment · Reviewer_th3c · 2024-11-27
> > > >
> > > > Thanks for the information. I'll keep my score unchanged.

---

### Official Review · Reviewer_CSd1 · 2024-10-30

**Soundness:** 3
**Presentation:** 1
**Contribution:** 2
**Rating:** 6
**Confidence:** 2

**Summary:**

The paper introduces a generalisation of the GP-LVM with the novel Q-Exp process. The training ELBO is derived and the performance is shown in multiple experiments to be improved upon GP-LVM.


After the first rebuttal answer, I have raised my score by one point.

**Strengths:**

Disclaimer: I am not familiar with QEP and am only partially familiar with GP-LVM. Thus my judgement is at times more superficial.

- The generalisation of GP-LVM offers more flexibility and improved performance
- While I have not managed to check all the mathematical details, they seem to be well executed.
- Experiments in small-scale toy problems demonstrate the performance of the method.

**Weaknesses:**

- The main weakness for me lies in the presentation which is hard to follow and at places unclear. See the questions below for more details on this point.
- Some claims such as the regularization effect and the connection to probabilistic PCA are not sufficiently worked out and need improvement.

**Questions:**

- Definition 1:
  - Can you give an intuition on how the q-exp distribution is different from the Gaussian potentially with examples? How does the distribution change with $q$?
  - What values can $q$ take in theory and what values are suitable in practice? You mainly focus on $0<q<2$ in the experiments, why is that?
- Remark 1: "more regularization". What form does the regularization take and where do I see this in the mathematical expressions? How does it connect to other commonly known forms of regularization?
- Section 2.2: can you clarify what the latent variables are here? With $Q\ll D$, I assume that $f$ is the latent but in (1) we have $y=f(x)+\varepsilon$. Then, in line 184 $X$ is the latent variable. This is confusing. Please clarify.
- Section 2: Can you give a brief overview on GP-LVM? This can help to compare your development more directly and see the differences.
- Line 161: How does replacing the GP with Q-EP impose more regularization?
- Remark 2:
  - How does $q>0$ regularize the singular values? Which singular values do you mean explicitly?
  - $c(q)$ is not defined?
- Figure 1:
  - What is the colour scale?
  - In the rightmost plot what is the artifact in the bottom left corner?
- Line 220: "can automatically determine the dimensionality of the nonlinear latent space". Can you explicitly clarify how this is achieved and show this in experiments? Currently, this claim is not fully supported.
- Line 261: "in two stages". Please clarify which stages.
- Line 264: What is $M$?
- Section 3.2.2:
  - The summary in section 3.2.2 is highly appreciated but can you additionally provide an algorithmic overview of how the model is trained and how it is used during inference?
  - What is the computational complexity of your method and how does it compare to GP-LVM? How does it scale larger sample sizes? What does this imply in practice for the use of your algorithm compared to GP-LVM? Please discuss.
  - What are the hyperparameters, e.g. $\beta$ and how are they chosen, what effect do they have, and how robust is the training towards those hyperparameters? Please discuss and if possible show in experiments.
- Remark 4: With $q=2$ you claim that the Gaussian case is recovered. But here you show that there is an extra term. What am I missing or where does this discrepancy arise from? This would affect all your experiments since then $q=2$ is not exactly the GP-LVM case anymore.
- Line 363: "superior" in what sense?
- Line 377: "best" in what sense?
- Line 377: "regularization effect", can you explicitly state how we can see this? Does this mean an axis alignment effect similar to Lasso regularization?
- Figure 2 (bottom): What is the x-axis of the right plots, $q$?
- Figure 3: Why is the 2d latent subspace so different with $q=1.57$ to the one in Figure 2 with $q=1.5$? How is the representation with $q=1.57$ good, the colours of the datapoint are overlapping?
- Table 2: The results of the Gaussian case are within the standard errors (at least in terms of AUC). Please highlight and discuss that.
- Line 528: The connection to probabilistic PCA is not made explicit. Please work that out to support your claim here and from the abstract.

Smaller details:
- Citations need to be revisited:
  - Line 42: Schö[l]kopf. Please check all citations for correctness.
  - Line 82: Lowercase the citation KLEPPE & SKAUG
  - E.g. Line 141: Citations that are used within the text should not contain brackets such as "(Li et al., 2023, in Theorem 3.5) prove...".
  - E.g. Shuyi Li, ... Bayesian Learning via q-exponential process, has the NeurIPs and arXiv reference.
- Wording needs improving in some places:
  - Line 161: "[The] GP..."
  - Line 411: "of [the] kernel"
- Introduce non-standard notation for easy understanding:
  - Line 179: $\bigotimes$
  - Line 196: $\wedge$
- Table 1/2: Why is the caption font size smaller?

---

> ### Author Response · Authors · 2024-11-20
> **Clarification on regularization effect and improvement of presentation**
>
> We thank you for your careful reading and appreciate your acknowledgment on the strengths of our paper. The numerical advantages of QEP-LVM have been supported by simulation and real data at different scales (from 3 dimensions to 784 dimensions).
>
> Weaknesses:
>
> * Regularization by parameter $q$ is explained in Remark 2 of Li et~al (2023). We realize that it may cause confusion without explicit mentioning. Therefore, we have added Remark 1 after Definition 1 to explain the connection to the common regularization by vector norm.
> * We have revised paragraphs in Section 3.1 to elaborate more on probabilistic PCA and its connection to LVM. Now the clarify has been improved.
>
> Questions:
>
> * Definition 1:
>   * Q-Exp family of distributions parameterized by $q>0$ includes Gaussian as a special case when $q=2$. For $0<q<2$, Q-ED is log-convex near 0 and heavy-tailed and when $q>2$, Q-ED is log-concave near 0 and light-tailed.
>   * In theory $q>0$ and you are right, interesting applications usually happen in $q\in (0,2]$. As commented in Section 4, the **regularization effect** on the latent representation refers to that smaller $q$ tends to contract the latent space. Usually we want to have more more compact representation than that by GP-LVM. Numerical experiments show that more interpretable latent representations are obtained for $q$ in this range. Those optimal $q^*$s picked by specific datasets are not restricted a priori in any region on the positive axis.
>
> * Remark 1 (now remark 2): the connection to commonly known forms of regularization has been clarified in Remark 1. Basically, it refers to the native log-density having the dominant term in the format of a weighted q-vector norm of the random variable (modeling parameter).
> * Section 2.2: latent variable is clearly stated on line 153. Note, latent function $f$ in the standard regression is different from the latent variable $X$ in LVM.
> * Section 2: The introduction has a detailed overview of GP-LVM. There is also a brief overview at the beginning of Section 3. Section 2 is saved for the introduction of Q-EP. The revised paragraphs in Section 3.1 has contained more details of GP-LVM. The key difference between QEP-LVM and GP-LVM is that the former replaces GP with Q-EP in the formulation of the latter.
> * Line 161: This should be explained by Remark 1 now and further illustrated by Figure 1 (previous Figure 2).
> * Remark 2 (now remark 3):
>   * Singular values refer to the those diagonal entries $\{l_i\}$ in the singular value decomposition of ${\bf X}$ in Theorem 3.1.
>   * $c(q)$ is the constant $c$, which is in turn a function of $q$, as show in the Theorem 3.1.
> * Figure 1 (now Figure A.1):
>   * The color scale is $[0,1]$. The color is only used for illustration, not for training.
>   * Yes, it is artifact caused by numerical resolution of plotting.
> * Line 220: it is actually quoted from Titsias \& Lawrence (2010). An explanation has been appended.
> * Line 261: two stages are from Hensman et al. (2015). It is not directly relevant so the sentence has now been rephrased to avoid confusion.
> * Line 264: ${\bf M}$ the mean of variational distribution for inducing values, part of the variational parameters to be optimized.
> * Section 3.2.2:
>   * The solution procedure for Bayesian QEP-LVM is the same as Bayesian GP-LVM, but with a more complicated ELBO as the loss function. The model is trained in \texttt{GPyTorch} using SGD.
>   * Good question. The computational complexity has now been added to Remark 5 (previous remark 4).
>   * kernel parameters, including $\beta$, is part of the parameters to be optimized in the standard SGD procedure as implemented in \texttt{PyTorch} and \texttt{GPyTorch}.
>
> * Remark 4 (now remark 5): the extra term comes from the application of Jensen's inequality and calculations that follow. It does not affect our experiments because when $q=2$ the code falls back to GP-LVM which is already implemented in \texttt{GPyTorch}.
> * Line 363: "superior" is in the sense of enhanced interpretability mentioned in that sentence. More advantages, including i) flexibility to regularize on the learned latent representations; ii) effective identification of the intrinsic latent dimensionality; iii) learning latent representations with better interpretability and adaptability for supervised/semi-supervised learning, have been demonstrated by various examples.
> * Line 377: among these plots.
> * Line 377: Good point. See above explanation.
> * Figure 2 (now Figure 1) bottom: x-axis refers to order of the inverse length-scale ${\bf \gamma}$.
> * Table 2: What do you mean by "within the standard errors"? The standard errors are based on 10 repeated experiments. Smaller values indicate that the mean results are robust to random initialization.
> * Line 528 (now 532): It is stated in Theorem 3.1.
>
> Smaller details: Fixed.
>
> All the relevant changes in the [**Rebuttal Revision**](/pdf/93b34ea5f98a2830c1c4afc11cc761fafe2430e5.pdf) are marked in $\textcolor{orange}{\text{orange}}$.

---

> > ### Comment · Reviewer_CSd1 · 2024-11-25
> > **Answer on rebuttal**
> >
> > Thank you for your clarifications. Many of my points have been answered. For some, I would urge the authors to add the answers to my questions additionally to the manuscript, e.g.
> > - first bullet point about Def 1,
> > - comment on line 264,
> > - comment about Remark 4,
> > - comment about Figure 2 (this is really important!).
> >
> > With my point about "within the standard errors", I mean that the Gaussian case has an AUC of $0.981\pm0.0062$ while the $q=1$ case has an AUC of $0.986\pm0.0065$. This means that the standard error intervals of both results overlap heavily, meaning that the AUC of the Gaussian case is not significantly worse than for $q=2$. Could you please discuss and clarify?
> >
> > Furthermore, while the references have been revised, additionally dois and URLs should be removed.
> >
> > I have raised my score accordingly, but I urge the authors to add your rebuttal clarifications, which are currently missing from the manuscript.

---

> ### Author Response · Authors · 2024-11-26
> **Thank you for raising your score**
>
> Thank you for your careful reading. we greatly appreciate that you have raised the score. We have modified the [Rebuttal Revision](/pdf/dff2db890d6680986794b21ea627f3431b4f5a3e.pdf) to add the edits as you mentioned. They are marked in $\textcolor{orange}{orange}$.
>
> Thank you for your clarification about "within the standard errors". You are right these standard error intervals may overlap, but QEP-LVM still has slightly better mean. AUC is one of the metrics we use (QEP-LVM does not fall behind). Other metrics firmly support the advantage of QEP-LVM compared with GP-LVM.

---

> > ### Comment · Reviewer_CSd1 · 2024-11-27
> > **Answer on rebuttal**
> >
> > Thank you for clarifying my last points.
> >
> > While I think that the paper has an interesting contribution, I am still slightly worried about the clarity of the presentation. This might be only because I am not entirely familiar with the literature. Thus, given the papers interestings contribution, I would like to stay with my raised borderline score of 6 to give the AC the chance to make a better informed decision about acceptance.

---

> > > ### Author Response · Authors · 2024-11-27
> > >
> > > Thank you very much for confirming again the contribution of our paper! Should you have any further questions, feel free to ask. We will be happy to try our best to answer them.

---

### Official Review · Reviewer_u49y · 2024-10-31

**Soundness:** 3
**Presentation:** 3
**Contribution:** 2
**Rating:** 5
**Confidence:** 3

**Summary:**

This paper proposes a model that extends the Gaussian Process Latent Variable Model (GP-LVM) to a non-Gaussian framework by employing the Q-exponential process (QEP). Specifically, the authors replace the Gaussian log-likelihood’s $ l_2 $ norm component with an $ l_q $ norm for $q > 0$, introducing an alternative to Gaussian assumptions. They derive both Maximum Likelihood Estimation (MLE) and variational Bayesian inference methods for this model. Experimental evaluations are conducted on the Swiss roll, oil flow, and MNIST datasets, and the results are compared with those of the GP-LVM.

**Strengths:**

- **Clarity**: The paper is well-organized and clearly written, making it accessible for readers.
- **Contextualization**: The authors provide a thorough review of related work.
- **Theoretical Derivation**: While I haven't exhaustively verified here, the paper appears to present a sound derivation of the ELBO within the variational Bayesian framework.

**Weaknesses:**

- **Significance of Contribution**: The experiments focus on datasets with limited scale and relatively artificial characteristics, which limits the persuasiveness of the findings. The results do not convincingly demonstrate a substantial improvement over the GP-LVM. Testing the proposed method on larger datasets or more practical tasks (e.g., robotics) could improve the impact of the results.
- **Comparison with Other Methods**: The experimental comparisons are restricted to the GP-LVM. To better assess the advantages of the QEP-LVM, it would be valuable to include comparisons with other methods suited for non-Gaussian or manifold data, such as Isomap for Swiss roll or neural networks for classification tasks like MNIST and oil flow.

**Questions:**

For more detailed points, please refer to the weaknesses section above.

Minor Comments:
- The notation $ q-ED $ could be improved as it might be misinterpreted as "q minus ED." A clearer notation, such as $ q_{ED} $, might enhance readability.

---

> ### Author Response · Authors · 2024-11-20
> **added robot learning example and VAE for comparison**
>
> We thank you for acknowledging the strengths of our paper. The motivation of the proposed QEP-LVM is not to replace GP-LVM, but rather to generalize it with flexibility on regularizing the learned latent representation. This, to our best knowledge, is novel.
>
> Weaknesses:
>
> * **Significance of Contribution**:
> The interesting findings in simulation and real data at different scales (from 3 dimensions to 784 dimensions) support QEP-LVM can i) regularize the learned latent representations; ii) effectively identify the intrinsic latent dimensionality; iii) learn latent representations with better interpretability and adaptability for supervised/semi-supervised learning. To further support these claims, we have added a robot learning example in Appendix C.2 where it is important to learn latent representation of motor motions to improve the performance of robots. QEP-LVM ($q=1.0$) is the only model that discovers the intrinsic dimensionality 3 (DOFs) of the latent space.
>
> * **Comparison with Other Methods**:
> We very much appreciate your suggestions on adding more comparisons. We have included VAE in the comparison throughout Section 4 since VAE is closely related to PCA. In general, we found that the latent representations learned by VAE is less structured and difficult to interpret. Note, both GP-LVM and QEP-LVM are unsupervised learning algorithms (class labels are not used). Tables only compare their derived generative classifiers (class labels used) to indirectly compare the quality of learned latent representations. Classification is not the main focus so It will be irrelevant and unfair to compare LVMs with neural networks for classification. Similarly, Isomap is a manifold learning algorithm, which is not our focus either. By adding VAE to the comparison on the latent representation learning, the numerical advantages of QEP-LVM have been better supported.
>
> Minor comments:
>
> Thanks. We have revised the notation to enhance readability.
>
> All the relevant changes in the [**Rebuttal Revision**](/pdf/93b34ea5f98a2830c1c4afc11cc761fafe2430e5.pdf) are marked in $\textcolor{blue}{\text{blue}}$.

---

> > ### Comment · Reviewer_u49y · 2024-11-27
> >
> > Thank you for your response. I appreciate that you conducted the experiment with the robotic data additionally. I read C.2. I feel the current results are bit weak to say that the proposed method is the best. Firstly, it would be better to use beta VAE instead of VAE. Beta VAE has a hyperparameter beta, which controls the tradeoff btw the intrinsic dimensionality and the reconstruction error. This beta should be chosen as q is chosen in your method. Secondly, how can we know 3 DoF is the correct intrinsic dimension? Thirdly, how can we numerically check whether the proposed method detects the true intrinsic dimension, say 3? Can we use something like a statistical test?
> >
> > In addition, I think manifold learning methods are quite relevant to your study. Compared with GPLVM, both are nonlinear and unsupervised. In addition, GP's kernel can be thought of as a distance of the observation space. In that sense, GPLVM learns the metric by adjusting the latent variables. Indeed Swissroll is a typical task of manifold learning. So it could be valid to compare with e.g. isomap.

---

> > > ### Author Response · Authors · 2024-11-27
> > >
> > > Thank you for taking time to read our newly added results and your follow-up comments. The dataset has no labels so we only compare their latent representations and essential latent dimensionality. If you are aware of any better quantitative comparison to make the results stronger, we would be happy to adopt.
> > >
> > > Thanks for bringing up beta-VAE. We do notice some interesting connection between beta-VAE and QEP-LVM. Following your suggestion, we have added the comparison with beta-VAE in to Figure C.5 (previous Figure C.4) in Section C.2. The new [**Rebuttal Revision**]((/pdf/f83f452cf5b18f2f61a6c49e7ee38d59ee070650.pdf) contains the updated results. Interestingly, beta-VAE also finds 3 intrinsic dimensions, as indicated by the number of dominant variances of latent distribution. The intrinsic dimensionality in this example is based on the fact that the dataset is constructed using 3 DOFs. Similarly, the Swiss roll/hole data have 2 intrinsic dimensions because of its 2d surface nature, though the data points sit in a 3d ambient space. We do not have any formal statistical test for intrinsic dimensionality. But we are open to any suggestions and appreciate any references.
> > >
> > > After a second thought, we agree with you that Isomap is related. So we have added Figure C.2 for latent representation by Isomap and t-SNE. Isomap unrolls the Swiss roll and the Swiss hole so its latent representation loses the rolling structure. T-SNE does not respect the continuous nature of the datasets.

---

> > > > ### Comment · Reviewer_u49y · 2024-11-28
> > > >
> > > > Thank you for conducting additional experiments. I think adding the comparison with beta-VAE definitely enhances the study. It seems beta-VAE is now competitive. What are the advantages of your method over beta-VAE?
> > > >
> > > > I understand the intrinsic dimension of the datasets.
> > > >
> > > > Unfortunately, I have no idea about quantitative evaluations.

---

> > > > > ### Author Response · Authors · 2024-11-28
> > > > >
> > > > > Thank you for your follow-up comments and questions.
> > > > >
> > > > > Per your request, we quickly added beta-VAE for comparison in this example. We are glad that our effort in such a short time could be appreciated. Beta-VAE happened to verify the intrinsic dimensionality of this dataset with QEP-LVM. However, as we commented, the latent representation by beta-VAE is disperse and unstructured as GP-LVM. From this point, it is hard to judge that beta-VAE is competitive. Based on our experiences with other projects, latent representation by many AE-type algorithms tends to be less interpretable.
> > > > >
> > > > > We want to emphasize our motivation and major contribution of the proposed work so we stay on our focus: QEP-LVM generalizes GP-LVM with innovative flexibility to regularize the learned latent representation through a tunable parameter $q$. When properly chosen, QEP-LVM could lead to better interpretability of latent representation and effective determination of intrinsic dimensionality, which has been well demonstrated and supported by numerical examples. Given the time constraint of rebuttal/discussion period, it is not very practical to chase after algorithms one another, not to mention that LVM and VAE are not a complete pear-to-pear comparison since the former does not have an encoding map. What we proposed in this work is a novel latent presentation learning algorithm with control on the regularization, which of course could be further improved by combining with new ideas.

---

### Official Review · Reviewer_mVse · 2024-11-04

**Soundness:** 3
**Presentation:** 3
**Contribution:** 2
**Rating:** 6
**Confidence:** 3

**Summary:**

This paper extends Gaussian Process Latent Variable Models (GP-LVM) by replacing the Gaussian Process with a more general Q-exponential process, thereby obtaining a series of new LVMs, Q-exponential Process LVMs (QEP-LVM). The author points out the relationship between QEP-LVM and non-linear probabilistic PCA, derives the ELBO of Bayesian QEP-LVM, and introduces a mechanism to optimize the ‘q’ of q-exponential distribution. Experiments prove the effectiveness of the proposed QEP-LVM.

**Strengths:**

1.	The motivation of this paper is intuitive and reasonable.
2.	The research on the proposed QEP-LVM is very solid. Not only does it reveal the relationship between QEP-LVM, GP-LVM and non-linear PCA, but it also derives the tractable ELBO of Bayesian QEP-LVM. In addition, a Bayesian approach is developed to obtain the optimal q.
3.	This paper is clearly written.

**Weaknesses:**

1.	The paper includes numerous visualization experiments; however, some, like Figure 5, do not reveal significant performance differences between QEL-LVM and GP-LVM. It is recommended to incorporate more numerical comparisons. For instance, for the cluster formed by the two numbers in Figure 5, utilize evaluation metrics from clustering algorithms to assess and compare the proposed QEP-LVM and GP-LVM.
2.	It appears that the proposed QEP-LVM requires additional training during the testing phase to compute the ELBO. How long does this training take, and will it impact the practicality of QEP-LVM?

**Questions:**

In the visualization of the latent representation in Figure 3, it seems that the separation of points is not as good as in Figure 2 when q=1.5. Is this an illusion? If not, what is the reason for it?

---

> ### Author Response · Authors · 2024-11-20
> **added more evaluation metrics**
>
> We thank you for acknowledging the strengths of our paper on the novelty and theoretic establishments.
>
> Weaknesses:
>
> 1. We realize that pairwise comparison may not emphasize enough the advantage of QEP-LVM over GP-LVM. Therefore, we have replaced it with 2d projections of latent variables for all classes by t-SNE. Seen from Figure 4 (previous Figure 5), unlike VAE generating unstructured latent representation with all digits mixed up, outputs by LVMs are much more interpretable. While GP-LVM has segregated clusters of digits 6 and 7 respectively, QEP-LVM has more concentrated clusters with each further apart from others. There lacks good metrics for direct evaluation of latent representation. Yet we did have quantitative comparison in Table 2 based on derived generative classifiers. As indirect evaluation, higher classification accuracy by QEP-LVM supports better latent representation on which the classifier is derived. Neither GP-LVM nor QEP-LVM is clustering algorithms because none outputs cluster assignments. However, we borrow the class labels (not used in training) as cluster assignments and have added two metrics for evaluating clustering algorithms, adjusted rand index (ARI) score, normalized mutual information (NMI) score, in the Table 2. QEP-LVM attains higher scores in both metrics.
> 2. The additional terms in ELBO (9) for QEP-LVM have the same computational complexity as other terms. Therefore, it does not significantly increase the computation compared with GP-LVM. We have added complexity in Remark 5 (previous remark 4) and running time in Tables. They illustrate that computational overhead is marginal.
>
> Questions:
>
> Thanks for pointing this out. We trained the model for longer time and have achieved better latent representation at $q^*=1.39$ instead of $q^*=1.57$, as illustrated in Figure 2 (previous Figure 3).
>
> All the relevant changes in the [**Rebuttal Revision**](/pdf/93b34ea5f98a2830c1c4afc11cc761fafe2430e5.pdf) are marked in $\textcolor{red}{\text{red}}$.

---

> ### Comment · Reviewer_mVse · 2024-11-27
>
> The authors have addressed my concerns. I'd like to increase the score to 6.

---

> > ### Author Response · Authors · 2024-12-02
> >
> > Thank you very much for raising your score! Your support is greatly appreciated!

---

### Author Response · Authors · 2024-11-20
**Global response**

We sincerely thank all the anonymous reviewers for their careful reading and constructive comments and suggestions. We have carefully addressed all the raised issues in the revised version, which has been uploaded to the **Rebuttal Revision** (can be [downloaded](/pdf/f83f452cf5b18f2f61a6c49e7ee38d59ee070650.pdf) from the same pdf link as original submission).

The main changes are as follows:
1. We now have added Variational Autoencoder (VAE) into comparison.
2. We have added an example of robot learning in Appendix C.2 and demonstrated that QEP-LVM ($q=1.0$) is the only model that discovers the intrinsic dimensionality of the latent space, indicated by the degree of freedoms (DOFs) of robot motion.
3. We have clarified the "regularization by parameter $q$" and the "regularization effect on latent representation learning".
4. We have revised the introduction of probabilistic PCA and its connection of GP-LVM.

Below are point-to-point responses. When revising the paper, we colored changes in response to different reviewers. Hopefully this facilitates the review and the discussion. We hope that the reviewers and the area chairs agree that the paper has been significantly improved to meet the high standards of ICLR. Thanks!

Authors

---

### Meta-Review · Area_Chair_Ehts · 2024-12-21

**Metareview:**

The authors propose a new method to regularize the latent representation through a tuneable parameter. The reviewer's are largely in agreement regarding the novel contributions of this work, while one reviewer raised concern about lack of enough empirical evidence. However, the flexibility that comes with the method being tuneable adds to the overall usefulness of the method, and I think the community would benefit from knowing about this in general.

**Additional Comments On Reviewer Discussion:**

During the discussion phase,  3 of the 4 reviewers voted for acceptance. the reviewer u49y raised a concern about how close the empirical results are with beta-vae. the authors have addressed this, highlighting the main contribution of the work being improving the tuneability of the latent representation intrinsic dimension, and the fact that beta-vae is not really a valid baseline for the proposed improvement. i favor the authors response on this.

---

### Decision · Program_Chairs · 2025-01-22

Accept (Poster)